# SpeechWakBench: How Well do Large Language Models Speak with Watermarks?

## Abstract

The rise of large language model (LLM)-based text-to-speech (TTS) synthesis has enabled unprecedented voice cloning capabilities, calling for robust content governance. In-processing watermarking, which embeds watermarks during generation, has proven effective for text and images. The immediate research question is to adapt in-processing watermarks to LLM-based TTS models, which similarly generate discrete tokens before synthesis Their transferability, in terms of quality and robustness, to speech remains a critical yet unverified conundrum. We present **SpeechWakBench**, the first large-scale benchmark to systematically evaluate the transferability of in-processing watermarking from LLMs to speech synthesis. **SpeechWakBench** evaluates 6 adapted in-processing LLM watermarking methods against 4 post-processing audio watermarking baselines across 3 modern LLM-based TTS models, using 16 reference-free quality metrics and a unified detectability metric under 10 attacks. Our results show that while in-processing watermarking produces slightly higher speech quality, it fails catastrophically in robustness, performing substantially worse than post-processing methods. We demonstrate that this failure is systemic, caused by the irreversible token-to-waveform conversion. This fundamental limitation highlights potential opportunities for developing novel watermarking approaches that are specifically tailored to address the unique challenges of speech synthesis. Our code is available at https://anonymous.4open.science/r/SpeechWakBench-1462.

## 1 Introduction

Large language model (LLM)-based text-to-speech (TTS) synthesis has reached a critical inflection point where generated speech achieves near-human perceptual quality and enables sophisticated capabilities such as zero-shot voice cloning from only a few seconds of reference audio (Du et al., 2024; Guo et al., 2024; Wang et al., 2025a). While these advances create opportunities in accessibility, entertainment, and human-computer interaction, they also introduce profound societal risks, including voice-based deepfakes and financial fraud through impersonation. These call for solutions, such as watermark, to avoid undermining the digital audio communications (Roman et al., 2024).

Highly realistic synthetic speech has created a pressing need for robust content governance mechanisms that can distinguish human speech from AI-generated audio. Unlike reactive deepfake detection systems that struggle to generalize across evolving generative models, watermarking provides a proactive defense by embedding signatures within the generation pipeline (Liu et al., 2024d). Most existing speech watermarks operate in a post-processing paradigm, which embeds watermarks into the final waveform after generation (Chen et al., 2023; Liu et al., 2024c; Roman et al., 2024; Singh et al., 2024). These methods typically involve additive modifications, which can cause quality degradation and noticeable distortions. In contrast, in-processing watermarks for text (Kirchenbauer et al., 2023; Dathathri et al., 2024) and image (Wen et al., 2023; Yang et al., 2024b) synthesis integrate into the generation process, showing a superior trade-off between imperceptibility and robustness over post-processing ones. This success raises a key question of whether the transferability of the in-processing paradigm to speech synthesis can overcome current limitations of quality degradation.

State-of-the-art (SOTA) LLM-based TTS models such as FireRedTTS (Guo et al., 2024), Fish-Speech (Liao et al., 2024), and Spark-TTS (Wang et al., 2025a) undergo a paradigm shift by employing a two-stage generation process. An LLM first autoregressively generates discrete speech

## SpeechWakBench

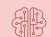 **Text-to-Speech Models**   FireRedTTS, Fish-Speech, Spark-TTS

**Watermarking Methods**
**In-processing**
KGW, Unigram, SWEET
MorphMark, SynthID, EXP
**Post-processing**
WavMark, Timbre
AudioSeal, SilentCipher

**Attacks**
Time stretch, Smooth
Gaussian noise, Background noise
Echo, MP3 compression
EnCodec, Quantization
High-pass filter, Low-pass filter

**Metrics**
**Quality**
WER, Speaker Similarity
UTMOS, PLCMOS, . . .
DNSMOS Overall, Squim PESQ
**Detectability**
TPR@X%FPR

Figure 1: SpeechWakBench evaluates 10 watermarking methods across 3 LLM-based TTS models, using 16 quality metrics and a unified detectability metric under 10 attack scenarios.

tokens from a learned codebook, which are then converted into audio by a separate vocoder. This intermediate token generation process is functionally equivalent to text generation in language models, creating an opportunity to adapt existing LLM watermarking techniques (Kirchenbauer et al., 2023; Liu et al., 2024b; Wu et al., 2023) to the speech domain.

We present **SpeechWakBench**, the first comprehensive benchmark to systematically evaluate the transferability of in-processing LLM watermarking techniques to speech synthesis. Our investigation includes 6 SOTA in-processing watermarks adapted for TTS architectures alongside 4 post-processing audio watermarking baselines across 3 LLM-based TTS models that support zero-shot voice cloning. We specifically introduce 16 reference-free quality metrics, unique to evaluating in-processing watermarks, and a unified detectability framework based on TPR@X%FPR to standardize evaluation. Our surprising findings reveal that successful LLM watermarking paradigms cannot be blindly transferred to speech. Contrary to expectations from text and image watermarking, in-processing speech watermarks show weaker robustness than post-processing approaches due to vulnerabilities introduced when reversing the token-to-waveform conversion. This finding challenges prevailing assumptions about the superiority of in-processing watermarking (Kirchenbauer et al., 2023; Dathathri et al., 2024) and highlights the need for watermarking methods designed specifically for LLM-based speech synthesis. To summarize, we list our contributions as follows:

1. We present **SpeechWakBench**, the first large-scale benchmark to systematically evaluate the transferability of in-processing watermarking from LLMs to speech synthesis.

2. We conduct the first systematic study of speech watermarking by comparing 6 in-processing and 4 post-processing methods across 3 SOTA LLM-based TTS models under 10 attacks.

3. We introduce a quality evaluation protocol, tailored to in-processing watermarks, using 16 reference-free metrics that eliminate bias from ground truth.

4. We establish a unified detectability assessment framework using TPR@X%FPR that standardizes evaluation across methods with different detection statistics, such as $p$-value or bit accuracy.

## 2 SPEECH SYNTHESIS AND WATERMARKS

We review related works on post-processing speech watermarks, in-processing LLM watermarks, and LLM-based TTS models. To the best of our knowledge, this is the first work to investigate in-processing watermarking for speech LLMs.

**Post-processing speech watermarks** embed watermarks after speech generation by modifying the final waveform. This process allows the use of non-watermarked reference speech for assessing quality degradation through reference-based metrics such as Signal-to-Noise Ratio (SNR) and perceptual distance measures. WavMark (Chen et al., 2023) introduces invertible neural networks for reciprocal encoding-decoding, while Timbre Watermarking (Liu et al., 2024c) targets voice cloning detection through frequency-temporal watermarking. More recent work like AudioSeal (Roman et al., 2024) uses jointly trained generator-detector networks optimized for real-time deployment, and SilentCipher (Singh et al., 2024) integrates psychoacoustic models and compression layers to

Table 1: Comparison of SpeechWakBench with existing works. Number of { IN In-processing Watermarks; POST Post-processing Watermarks; ATT Attacks; RD Real Datasets; SD Synthetic Datasets; REF-B Reference-based Quality Metrics; REF-F Reference-free Quality Metrics}.

| Research Work | IN | POST | ATT | RD | SD | REF-B | REF-F | Detectability Metric |
|---|---|---|---|---|---|---|---|---|
| WavMark[1] | ✗ | 4 | 10 | 4 | ✗ | 1 | 1 | Bit accuracy |
| Timbre[2] | ✗ | 4 | 15 | 2 | ✗ | 1 | 3 | Bit accuracy |
| AudioSeal[3] | ✗ | 2 | 14 | 2 | 4 | 1 | 3 | Bit accuracy, TPR, FPR |
| SilentCipher[4] | ✗ | 4 | 11 | 3 | ✗ | 1 | ✗ | Bit accuracy |
| AudioMarkBench[5] | ✗ | 3 | 15 | 2 | ✗ | 2 | ✗ | Bit accuracy, FNR, FPR |
| **SpeechWakBench** | 6 | 4 | 10 | ✗ | 2 × 3 (models) | ✗ | 16 | TPR@X%FPR |

[1] Chen et al. (2023). [2] Liu et al. (2024c). [3] Roman et al. (2024). [4] Singh et al. (2024). [5] Liu et al. (2024d).

preserve audio quality. However, these methods remain fragile against common distortions and neural transformations (Liu et al., 2024d). While AudioMarkBench (Liu et al., 2024d) provides a broad robustness evaluation, its reliance on reference-based metrics such as ViSQOL and SNR limits its applicability to in-processing methods, where no non-watermarked reference speech exists.

**In-processing LLM watermarks** integrate watermarking directly into the token generation process. Early methods such as KGW (Kirchenbauer et al., 2023) introduced the "green-list" paradigm, later extended by Unigram (Zhao et al., 2024) and SWEET (Lee et al., 2024). Production-ready watermarking systems like SynthID-Text (Dathathri et al., 2024) demonstrate deployment readiness with multi-state detection, while MorphMark (Wang et al., 2025b) highlights adaptability by dynamically selecting strategies. Semantic watermarking schemes such as SIR (Liu et al., 2024b), X-SIR (He et al., 2024), and k-SemStamp (Hou et al., 2024) embed meaning-level signals to withstand paraphrasing and cross-lingual transformations. Distribution-preserving methods like DiPmark (Wu et al., 2023), Unbiased Watermark (Hu et al., 2024), as well as noise-based approaches such as EXP (Aaronson & Kirchner, 2022) and Permute-and-Flip (Zhao et al., 2025), expand the design space by embedding watermarks while maintaining fidelity. Trust and verification have also been explored through frameworks such as UPV (Liu et al., 2024a). However, all of these methods have been developed for text generation, while their transferability to speech remains entirely unexplored.

**LLM-based TTS models** generate speech using a discrete tokenization strategy, where an LLM predicts speech tokens that a vocoder then renders into audio. In this paper, we focus on LLM-based TTS models that support zero-shot voice cloning. FireRedTTS (Guo et al., 2024) uses HuBERT-based tokenization with a flow-matching vocoder for deployment, Fish-Speech (Liao et al., 2024) employs grouped quantization within a dual autoregressive framework for multilingual synthesis, and Spark-TTS (Wang et al., 2025a) introduces a streamlined decoder-only architecture with Bi-Codec tokenization. These models achieve highly natural, expressive, and controllable audio with advanced capabilities such as zero-shot voice cloning and multilingual support. Importantly, their token prediction stage is architecturally equivalent to text generation with LLMs, making it possible to adapt in-processing LLM watermarking from text to speech.

## 3 SPEECHWAKBENCH

As shown in Table 1, SpeechWakBench stands out by covering 10 watermarking methods across 6 synthetic datasets generated from 3 LLM-based TTS models. It further incorporates 16 reference-free quality metrics and a unified detectability metric, making it the most comprehensive and realistic evaluation framework for speech watermarking to date. Following Figure 2, we provide an overview of SpeechWakBench, including the in-processing and post-processing pipeline. The process is mainly divided into two stages: (i) watermark embedding and (ii) watermark detection. Key notations are summarized in Appendix A. Details of benchmark design are provided in Appendix B.

### 3.1 IN-PROCESSING WATERMARKING ON LLM-BASED TTS MODELS

SOTA LLM-based TTS models such as FireRedTTS (Guo et al., 2024), Fish-Speech (Liao et al., 2024), and Spark-TTS (Wang et al., 2025a) follow a unified token-based generation pipeline that can be decomposed into two main stages.

Figure 2: Pipeline of SpeechWakBench. In-processing methods embed watermarks during token generation. For example, KGW watermark divides tokens into red and green lists. During token generation, more tokens are selected from the green list. Post-processing methods modify the waveform directly, encoding bits as black ("0") or white ("1") regions. Detection uses $p$-value for in-processing and bit accuracy for post-processing. All results are unified using TPR@X%FPR.

In the first stage, given input text $\mathcal{T}$, reference audio $\mathcal{A}_{\text{ref}}$ for zero-shot synthesis, and optional control attributes $\mathcal{C} = \{c_{\text{speed}}, c_{\text{pitch}}, c_{\text{emotion}}\}$, an autoregressive language model $\mathcal{M}_{\text{LLM}}$ generates a sequence of discrete speech tokens $\mathbf{x} = (x_1, x_2, \dots, x_T) \in \mathcal{V}^T$, where $\mathcal{V}$ denotes the discrete token vocabulary. The generation process follows:

$$p(x_t | x_{<t}, \mathcal{T}, \mathcal{A}_{\text{ref}}, \mathcal{C}) = \text{softmax}(\mathcal{M}_{\text{LLM}}(x_{<t}, \mathcal{T}, \mathcal{A}_{\text{ref}}, \mathcal{C})),$$

where $t$ denotes the time step in the autoregressive generation process. In Spark-TTS, this includes both semantic tokens $\mathbf{x}^{(s)} \in \mathcal{V}_s^{T_s}$ capturing linguistic content and global tokens $\mathbf{x}^{(g)} \in \mathcal{V}_g^{T_g}$ encoding speaker characteristics. In the second stage, the generated discrete tokens are decoded into waveform audio using a neural decoder $\mathbf{y} = \mathcal{D}(\mathbf{x}) \in \mathbb{R}^N$, where $\mathcal{D} : \mathcal{V}^T \to \mathbb{R}^N$ represents the neural vocoder.

### 3.1.1 WATERMARK EMBEDDING

In-processing watermarks modify the token generation process during the first stage using a secret key $k \in \mathcal{K}$. They can be grouped into two categories:

**Logit Modification Method**  Directly modify the probability distributions before sampling:

$$\tilde{\mathbf{l}}_t = \mathcal{W}_{\text{logit}}(k, \mathbf{l}_t, x_{<t}, t),$$

where $\mathbf{l}_t \in \mathbb{R}^{|\mathcal{V}|}$ are the original logits and $\tilde{\mathbf{l}}_t$ are the watermarked logits. For instance, KGW (Kirchenbauer et al., 2023) watermark generates a context-dependent secret key $k^{(t)} = H(x_{t-h}, \dots, x_{t-1}, k)$, where $H$ is a cryptographic hash function and $h$ is the context window size. This key seeds a random number generator to partition the vocabulary into disjoint sets $\mathcal{V} = \mathcal{G}_{k^{(t)}} \cup \overline{\mathcal{G}}_{k^{(t)}}$, where the green-list $\mathcal{G}_{k^{(t)}}$ contains $\gamma|\mathcal{V}|$ tokens for $\gamma \in [0, 1]$. The logit modification follows:

$$\tilde{\mathbf{l}}_t[v] = \begin{cases} \mathbf{l}_t[v] + \delta & \text{if } v \in \mathcal{G}_{k^{(t)}} \\ \mathbf{l}_t[v] & \text{if } v \in \overline{\mathcal{G}}_{k^{(t)}} \end{cases},$$

where $\delta > 0$ controls watermark strength. After softmax normalization, this increases green-list token probabilities while maintaining $\mathbb{E}_k[\tilde{p}_t] = p_t$ across different keys.

**Sampling Modification Method**  Alter the sampling strategy while preserving the original logits:

$$x_t \sim \mathcal{P}_{\text{watermark}}(k, \mathbf{l}_t, x_{<t}, t),$$

where $\mathcal{P}_{\text{watermark}}$ incorporates watermark patterns through deterministic processes. For example, EXP (Aaronson & Kirchner, 2022) watermark generates a secret vector $\mathbf{r}^{(t)} \in [0, 1]^{|\mathcal{V}|}$ using the same context-dependent key $k^{(t)} = H(x_{t-h}, \dots, x_{t-1}, k)$. For each token $v$, a uniform random value $r_v^{(t)} \sim \text{Uniform}(0, 1)$ is sampled from the seeded generator. The token selection follows the exponential minimum principle:

$$x_t = \arg\max_{v \in \mathcal{V}} \left\{ (r_v^{(t)})^{1/p_t[v]} \right\},$$

where $p_t[v] = \text{softmax}(\mathbf{l}_t)[v]$. This deterministic selection maintains the exact original distribution $P(x_t = v) = p_t[v]$ over the randomness in the secret vector. Logit modification methods create a direct trade-off between detectability and text quality, as a stronger signal requires a larger distortion of the original probability distribution, potentially leading to lower quality generated tokens. In contrast, some sampling modification methods (Kuditipudi et al., 2024) aim to be "distortion-free" by preserving the original distribution, but this can weaken the detectability of the watermark.

### 3.1.2 WATERMARK DETECTION

Watermark detection requires the inversion of the generation pipeline, i.e., from speech to tokens, and applying statistical hypothesis testing. The process involves three main steps:

1. **Token Recovery**   Extract discrete semantic tokens from potentially watermarked speech $\hat{\mathbf{y}} \in \mathbb{R}^N$ using the speech encoder:
$$\hat{\mathbf{x}} = \mathcal{E}(\hat{\mathbf{y}}) \in \mathcal{V}^T,$$
   where $\mathcal{E} : \mathbb{R}^N \to \mathcal{V}^T$ is the speech encoder (e.g., BiCodec's semantic tokenizer).

2. **Statistical Hypothesis Testing**   Test the hypothesis $H_0$: "the semantic tokens are without watermark" against $H_1$: "the semantic tokens are watermarked", then compute a detection score $S_T$ based on the recovered token sequence $\hat{\mathbf{x}}$ and the secret key $k$
$$S_T = \mathcal{F}_{\text{score}}(k, \hat{\mathbf{x}}),$$
   where $\mathcal{F}_{\text{score}}$ is the scoring function that measures the statistical bias toward favorable tokens in the recovered sequence.

3. **Watermark Detection**   Calculate the $p$-value based on the score's distribution under $H_0$ and compare against a predetermined false positive rate $\alpha$. The speech is flagged as watermarked if p-value$(S_T) < \alpha$.

This detection process leverages the discrete nature of the LLM part of TTS models. Based on the prior performance of in-processing LLM watermarks on text, one would expect reliable watermark detection with theoretical guarantees on false positive rates without requiring access to the original generation model parameters.

### 3.2 BENCHMARK DESIGN AND EVALUATION PROTOCOL

**LLM-based TTS Models**   We evaluate watermarking methods on 3 recent LLM-based zero-shot TTS models: FireRedTTS (Guo et al., 2024), Fish-Speech (Liao et al., 2024), and Spark-TTS (Wang et al., 2025a). These SOTA models share a common architecture of discrete token prediction followed by neural vocoding, making them directly compatible with in-processing watermarking. At the same time, they differ in training data and architectural design, providing diversity in linguistic coverage and synthesis quality. Since watermark embedding requires access to model internals, our analysis focuses on open-source models. Nonetheless, the benchmark remains applicable for internal evaluation of closed-source models by model developers.

**Datasets**   To ensure a robust and comprehensive evaluation, we use 2 distinct benchmarks: Seed-TTS-Eval (Anastassiou et al., 2024) and CV3-Eval (Du et al., 2025). Seed-TTS-Eval is an out-of-domain test set specifically designed to assess zero-shot speech generation capabilities. It comprises English and Chinese samples drawn from public corpora, including 1,000 samples from Common Voice (Ardila et al., 2020) and 2,000 from DiDiSpeech-2 (Guo et al., 2021). Complementing this, the CV3-Eval benchmark addresses the limitations of traditional clean audiobook datasets like LibriSpeech (Panayotov et al., 2015) by including noisy and real-world recordings. This provides a more challenging evaluation of multilingual voice cloning and emotional expressiveness (Du et al., 2025). For our experiments, we use the English (EN) and Chinese (ZH) subsets from both benchmarks and generate speech samples using all 3 LLM-based TTS models. We perform zero-shot voice cloning by following the original speech and prompt from each dataset.

**Watermarking Methods**   In our benchmark, we evaluate 6 in-processing and 4 post-processing watermarking methods. The in-processing methods include KGW (Kirchenbauer et al., 2023), Unigram (Zhao et al., 2024), SWEET (Lee et al., 2024), MorphMark (Wang et al., 2025b), Google's

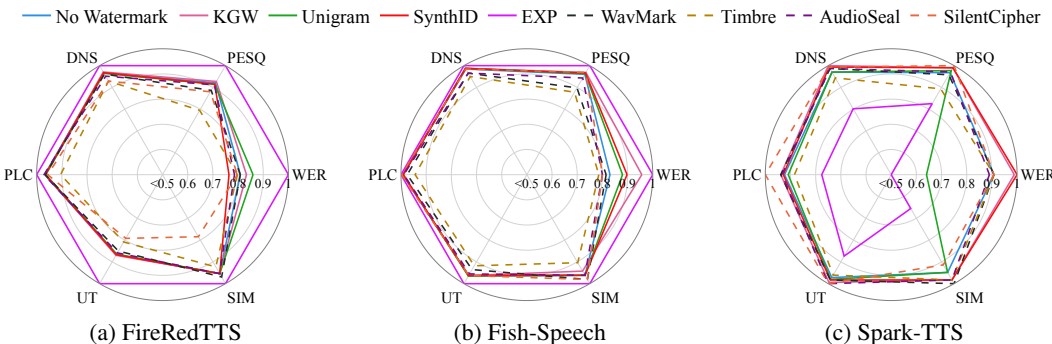

Figure 3: Quality results on CV3-Eval English dataset. All metrics are normalized to the percentage of the best performance per metric. WER is inverted for consistent interpretation.

SynthID (Dathathri et al., 2024), and EXP (Aaronson & Kirchner, 2022). On the other hand, we consider WavMark (Chen et al., 2023), Timbre (Liu et al., 2024c), AudioSeal (Roman et al., 2024), and SilentCipher (Singh et al., 2024) for post-processing methods.

**Attacks**   To assess robustness, we evaluate watermarking methods under 10 attack scenarios across multiple categories based on the implementation from AudioMarkBench (Liu et al., 2024d). Temporal distortions include time stretching (TS) and smoothing (SMH). Noise-based distorts cover Gaussian noise (GN), background noise (BN), and echo. Compression and quantization distortions consist of MP3 compression (MP3), EnCodec (ECD), and quantization (QNT). Finally, filtering distortions include high-pass filtering (HPF) and low-pass filtering (LPF). Parameter settings for each attack are provided in Appendix B.4.

**Quality Metrics**   A key challenge in evaluating in-processing watermarks is the absence of ground truth reference speech, which makes traditional reference-based metrics unsuitable. While post-processing watermarks can be assessed against the non-watermarked ones, this inherently biases the comparison. To ensure fairness across both paradigms, we adopt 16 reference-free metrics to assess the quality of watermarked or non-watermarked synthetic speech by following the VERSA benchmark (Shi et al., 2025). For the main analysis, we provide 6 of them according to prior works (Du et al., 2025; Wang et al., 2025a). These include Word Error Rate (WER) (Anastassiou et al., 2024) for intelligibility, Speaker Similarity (SIM) (Jung et al., 2024) for voice identity preservation, and several non-intrusive predictors of overall naturalness, including UTMOS (UT) (Saeki et al., 2022), PLCMOS (PLC) (Diener et al., 2023), DNSMOS Overall (DNS) (Reddy et al., 2022), and Torch-Squim PESQ (PESQ) (Kumar et al., 2023). All 16 metrics are shown in Appendix B.5.

**Unified Detectability Metric**   Different watermarking schemes can produce different detection outputs, such as $p$-values, bit accuracy, or empirical scores, which makes direct comparison difficult. To standardize evaluation, we adopt the True Positive Rate at a fixed False Positive Rate (TPR@X%FPR) as a unified detectability metric. This provides a consistent measure of detection reliability at a specified tolerance for false alarms (e.g., X = 3.0% or 0.2%). For $p$-value based methods, TPR is computed as the proportion of watermarked samples with $p$-value smaller than the chosen FPR threshold $\alpha$ (e.g., TPR@3.0%FPR uses $\alpha = 0.03$). For bit accuracy methods, we treat bit accuracy as a test statistic under the null hypothesis $H_0$ : "speech is non-watermarked" (random bit recovery), where for an $m$-bit message, $X \sim \text{Binomial}(m, 0.5)$ represents correctly recovered bits. To achieve X% FPR, the threshold is set as $\tau = \min\{k : P(\text{Binomial}(m, 0.5) \geq k) \leq \text{X}\%\}$. For example, with $m = 16$ bits, the thresholds are $\tau_{3.0\%} = 13$ (bit accuracy $\geq 81.25\%$) and $\tau_{0.2\%} = 14$ (bit accuracy $\geq 87.5\%$). For empirical score methods, we collect scores from $N$ non-watermarked samples and set the threshold as the $(100-\text{X})$-th percentile $\tau = \text{percentile}_{100-\text{X}}(\{s_1, s_2, \ldots, s_N\})$.

## 4   BENCHMARKING RESULTS AND ANALYSIS

We conduct extensive evaluations of speech watermarking across 3 aspects: quality, detectability, and robustness. Additional experimental results and extended analyses are included in Appendix C.

Table 2: Quality and detectability performance of watermarking methods on Seed-TTS-Eval.

| Model | Method | EN | | | | | | | | ZH | | | | | | | |
|---|---|---|---|---|---|---|---|---|---|---|---|---|---|---|---|---|---|
| | | Quality Metric | | | | | | TPR@X%FPR | | Quality Metric | | | | | | TPR@X%FPR | |
| | | WER ↓ | SIM ↑ | UT ↑ | PLC ↑ | DNS ↑ | PESQ ↑ | 0.2% ↑ | 3.0% ↑ | WER ↓ | SIM ↑ | UT ↑ | PLC ↑ | DNS ↑ | PESQ ↑ | 0.2% ↑ | 3.0% ↑ |
| FireRedTTS | No Watermark | 2.5 | 0.63 | 3.64 | 4.12 | 3.04 | 3.20 | — | — | 1.2 | 0.74 | 2.93 | 4.17 | 3.16 | 3.47 | — | — |
| | KGW | 2.6 | 0.63 | 3.62 | 4.10 | 3.04 | 3.20 | 0.875 | 0.966 | 1.1 | 0.75 | 2.92 | 4.16 | 3.15 | 3.48 | 0.997 | 1.000 |
| | Unigram | 2.8 | 0.63 | 3.62 | 4.12 | 3.05 | 3.18 | 1.000 | 1.000 | 1.2 | 0.74 | 2.91 | 4.16 | 3.15 | 3.47 | 1.000 | 1.000 |
| | SWEET | 2.6 | 0.63 | 3.62 | 4.12 | 3.04 | 3.18 | 0.873 | 0.976 | 1.2 | 0.75 | 2.92 | 4.16 | 3.15 | 3.48 | 0.995 | 1.000 |
| | MorphMark | 2.5 | 0.63 | 3.63 | 4.13 | 3.04 | 3.18 | 0.062 | 0.274 | 1.2 | 0.74 | 2.93 | 4.17 | 3.16 | 3.47 | 0.101 | 0.360 |
| | SynthID | 2.6 | 0.63 | 3.61 | 4.11 | 3.04 | 3.19 | 0.966 | 0.995 | 1.2 | 0.74 | 2.92 | 4.15 | 3.15 | 3.47 | 0.998 | 1.000 |
| | EXP | 2.2 | 0.65 | 3.97 | 4.24 | 3.12 | 3.42 | 1.000 | 1.000 | 1.0 | 0.75 | 3.29 | 4.31 | 3.22 | 3.66 | 1.000 | 1.000 |
| | WavMark | 2.6 | 0.63 | 3.57 | 4.12 | 3.00 | 3.11 | 1.000 | 1.000 | 1.1 | 0.73 | 2.89 | 4.25 | 3.13 | 3.36 | 1.000 | 1.000 |
| | Timbre | 2.6 | 0.62 | 3.47 | 3.82 | 2.88 | 2.84 | 1.000 | 1.000 | 1.2 | 0.73 | 2.74 | 3.81 | 2.99 | 2.98 | 1.000 | 1.000 |
| | AudioSeal | 2.6 | 0.63 | 3.61 | 4.14 | 3.02 | 3.18 | 1.000 | 1.000 | 1.2 | 0.74 | 2.89 | 4.15 | 3.13 | 3.45 | 1.000 | 1.000 |
| | SilentCipher | 2.5 | 0.63 | 3.63 | 4.26 | 3.04 | 3.22 | 0.980 | 0.981 | 1.2 | 0.74 | 2.94 | 4.32 | 3.16 | 3.49 | 0.998 | 0.998 |
| Fish-Speech | No Watermark | 2.0 | 0.53 | 4.15 | 4.45 | 3.22 | 3.58 | — | — | 1.1 | 0.69 | 3.50 | 4.47 | 3.26 | 3.56 | — | — |
| | KGW | 1.8 | 0.53 | 4.14 | 4.44 | 3.22 | 3.58 | 0.075 | 0.235 | 1.2 | 0.68 | 3.45 | 4.47 | 3.25 | 3.56 | 0.253 | 0.515 |
| | Unigram | 2.3 | 0.53 | 4.14 | 4.44 | 3.22 | 3.58 | 1.000 | 1.000 | 1.2 | 0.69 | 3.46 | 4.46 | 3.25 | 3.55 | 0.996 | 0.998 |
| | SWEET | 2.0 | 0.53 | 4.14 | 4.44 | 3.22 | 3.59 | 0.066 | 0.243 | 1.3 | 0.68 | 3.45 | 4.46 | 3.25 | 3.56 | 0.289 | 0.544 |
| | MorphMark | 2.0 | 0.53 | 4.15 | 4.44 | 3.22 | 3.59 | 0.000 | 0.001 | 1.3 | 0.69 | 3.48 | 4.47 | 3.26 | 3.57 | 0.000 | 0.001 |
| | SynthID | 2.0 | 0.53 | 4.14 | 4.44 | 3.22 | 3.59 | 0.458 | 0.818 | 1.2 | 0.68 | 3.45 | 4.46 | 3.25 | 3.56 | 0.840 | 0.957 |
| | EXP | 2.1 | 0.54 | 4.21 | 4.46 | 3.24 | 3.66 | 1.000 | 1.000 | 1.2 | 0.69 | 3.54 | 4.49 | 3.28 | 3.61 | 0.998 | 0.999 |
| | WavMark | 2.1 | 0.55 | 4.04 | 4.46 | 3.16 | 3.38 | 1.000 | 1.000 | 1.2 | 0.67 | 3.39 | 4.52 | 3.18 | 3.35 | 1.000 | 1.000 |
| | Timbre | 2.1 | 0.51 | 4.04 | 4.32 | 3.14 | 3.39 | 1.000 | 1.000 | 1.2 | 0.67 | 3.32 | 4.30 | 3.18 | 3.25 | 1.000 | 1.000 |
| | AudioSeal | 2.1 | 0.53 | 4.14 | 4.45 | 3.19 | 3.55 | 1.000 | 1.000 | 1.2 | 0.69 | 3.46 | 4.46 | 3.21 | 3.53 | 1.000 | 1.000 |
| | SilentCipher | 2.1 | 0.54 | 4.15 | 4.45 | 3.22 | 3.60 | 0.969 | 0.969 | 1.2 | 0.69 | 3.50 | 4.47 | 3.27 | 3.57 | 1.000 | 1.000 |
| Spark-TTS | No Watermark | 2.7 | 0.59 | 3.93 | 4.39 | 3.12 | 3.31 | — | — | 1.6 | 0.67 | 3.28 | 4.38 | 3.22 | 3.58 | — | — |
| | KGW | 3.0 | 0.59 | 3.89 | 4.37 | 3.12 | 3.27 | 0.450 | 0.685 | 1.5 | 0.67 | 3.24 | 4.37 | 3.22 | 3.58 | 0.482 | 0.765 |
| | Unigram | 5.2 | 0.58 | 3.88 | 4.32 | 3.08 | 3.24 | 0.851 | 0.926 | 3.6 | 0.65 | 3.36 | 4.31 | 3.18 | 3.57 | 0.964 | 0.991 |
| | SWEET | 4.3 | 0.58 | 3.90 | 4.35 | 3.11 | 3.26 | 0.463 | 0.680 | 2.7 | 0.67 | 3.23 | 4.36 | 3.21 | 3.56 | 0.535 | 0.765 |
| | MorphMark | 2.6 | 0.59 | 3.94 | 4.39 | 3.14 | 3.31 | 0.032 | 0.182 | 2.7 | 0.67 | 3.27 | 4.37 | 3.22 | 3.59 | 0.044 | 0.224 |
| | SynthID | 2.8 | 0.60 | 3.90 | 4.38 | 3.13 | 3.30 | 0.365 | 0.626 | 1.4 | 0.67 | 3.23 | 4.36 | 3.22 | 3.57 | 0.000 | 0.778 |
| | EXP | 18.3 | 0.48 | 3.73 | 3.99 | 2.76 | 2.96 | 0.662 | 0.744 | 14.0 | 0.60 | 3.14 | 4.17 | 3.02 | 3.39 | 0.895 | 0.919 |
| | WavMark | 3.9 | 0.60 | 3.76 | 4.30 | 3.09 | 3.24 | 0.997 | 0.997 | 1.6 | 0.67 | 3.15 | 4.32 | 3.12 | 3.39 | 0.996 | 0.996 |
| | Timbre | 2.8 | 0.59 | 3.78 | 4.19 | 2.96 | 2.95 | 1.000 | 1.000 | 2.3 | 0.65 | 3.09 | 4.17 | 3.02 | 3.13 | 1.000 | 1.000 |
| | AudioSeal | 2.8 | 0.59 | 3.91 | 4.41 | 3.11 | 3.25 | 1.000 | 1.000 | 2.3 | 0.67 | 3.25 | 4.36 | 3.20 | 3.54 | 1.000 | 1.000 |
| | SilentCipher | 2.8 | 0.57 | 3.93 | 4.40 | 3.11 | 3.33 | 0.962 | 0.963 | 1.4 | 0.67 | 3.23 | 4.36 | 3.22 | 3.57 | 0.995 | 0.995 |

In-processing.    Post-processing.    ↑ Higher is better.    ↓ Lower is better.

## 4.1 BENCHMARKING SYNTHETIC SPEECH QUALITY AND WATERMARK DETECTABILITY

Table 2 shows that in-processing watermarking methods consistently outperform post-processing baselines. This indicates that embedding watermarks during token generation does not degrade the speech quality. As depicted in Figure 3, we observe that most post-processing methods (dashed lines) fall within the performance boundaries of in-processing methods (solid lines). Notably, the EXP watermark consistently achieves the best overall quality with FireRedTTS and Fish-Speech models. However, it struggles with Spark-TTS. This is because Spark-TTS fails to generate speech for certain input prompts, resulting in silent outputs. This limitation highlights the importance of considering architectures of LLM-based TTS models when applying in-processing watermarking.

To evaluate the watermark detectability, we report TPR@X%FPR using 0.2% and 3.0% thresholds, which correspond to $14/16$ and $13/16$ bit accuracy for a 16 bits watermark message. As shown in Table 2, post-processing baselines achieve a nearly perfect detection even at the stricter 0.2% FPR, whereas in-processing methods show an inconsistent and often weak performance. For instance, Unigram and EXP watermarks on FireRedTTS and Fish-Speech, and SynthID on FireRedTTS, achieve an almost perfect TPR@0.2%FPR. However, KGW and MorphMark watermarks underperform other baselines, especially on Fish-Speech model. This differs greatly from text (Kirchenbauer et al., 2023) and image (Yang et al., 2024b) watermarking, where in-processing methods are known to provide highly reliable detection. Additional results are provided in Appendix C.

## 4.2 BENCHMARKING WATERMARK ROBUSTNESS AGAINST ATTACKS

Table 3 presents robustness results under a wide range of speech transformations. Post-processing watermarks, except SilentCipher, successfully defend against more than half of the attacks. In contrast, in-processing methods are less reliable. Although the EXP watermark achieves strong detectability without attacks, it does poorly under most attacks, achieving TPR@0.2%FPR above 0.7 in only two cases on the Fish-Speech model, hence reflecting difficulties in balancing the trade-off between quality and robustness. Besides, almost all methods fail under EnCodec, quantization, and high-pass filtering attacks. This pattern is also consistently observed across our comprehensive attack evaluation reported in Appendix C, indicating fundamental limitations in the robustness of in-processing watermarking against realistic speech distortions.

Table 3: Robustness evaluation results on Seed-TTS-Eval under attacks. All metrics represent TPR@0.2% FPR (higher is better).

| Model | Method | EN | | | | | | | | | | ZH | | | | | | | | | |
|---|---|---|---|---|---|---|---|---|---|---|---|---|---|---|---|---|---|---|---|---|---|
| | | TS | SMH | GN | BN | Echo | MP3 | ECD | QNT | HPF | LPF | TS | SMH | GN | BN | Echo | MP3 | ECD | QNT | HPF | LPF |
| FireRedTTS | KGW | 0.000 | 0.006 | 0.149 | 0.270 | 0.000 | 0.824 | 0.020 | 0.008 | 0.000 | 0.875 | 0.000 | 0.006 | 0.602 | 0.788 | 0.000 | 0.995 | 0.032 | 0.033 | 0.000 | 0.997 |
| | Unigram | 0.092 | 0.051 | 0.742 | 0.870 | 0.015 | 0.998 | 0.369 | 0.086 | 1.000 | 1.000 | 0.203 | 0.142 | 0.991 | 1.000 | 0.143 | 1.000 | 0.727 | 0.310 | 1.000 | 1.000 |
| | SWEET | 0.000 | 0.002 | 0.151 | 0.255 | 0.000 | 0.844 | 0.017 | 0.014 | 0.000 | 0.873 | 0.000 | 0.003 | 0.584 | 0.772 | 0.002 | 0.995 | 0.037 | 0.021 | 0.000 | 0.995 |
| | MorphMark | 0.000 | 0.001 | 0.006 | 0.011 | 0.000 | 0.052 | 0.002 | 0.008 | 0.000 | 0.062 | 0.000 | 0.003 | 0.021 | 0.021 | 0.000 | 0.090 | 0.005 | 0.012 | 0.000 | 0.101 |
| | SynthID | 0.005 | 0.005 | 0.206 | 0.342 | 0.033 | 0.943 | 0.017 | 0.009 | 0.000 | 0.966 | 0.012 | 0.009 | 0.538 | 0.755 | 0.052 | 0.998 | 0.042 | 0.026 | 0.000 | 0.998 |
| | EXP | 0.050 | 0.097 | 0.919 | 0.987 | 0.342 | 1.000 | 0.591 | 0.092 | 0.000 | 1.000 | 0.049 | 0.110 | 0.998 | 1.000 | 0.534 | 1.000 | 0.622 | 0.316 | 0.000 | 1.000 |
| | WavMark | 0.506 | 1.000 | 0.987 | 1.000 | 0.976 | 1.000 | 0.000 | 0.159 | 0.000 | 1.000 | 0.770 | 1.000 | 1.000 | 1.000 | 1.000 | 1.000 | 0.000 | 0.408 | 0.000 | 1.000 |
| | Timbre | 1.000 | 1.000 | 1.000 | 1.000 | 0.999 | 1.000 | 0.523 | 0.998 | 0.000 | 1.000 | 1.000 | 1.000 | 1.000 | 1.000 | 1.000 | 1.000 | 0.451 | 1.000 | 0.000 | 1.000 |
| | AudioSeal | 0.993 | 0.999 | 1.000 | 1.000 | 0.999 | 1.000 | 0.829 | 0.988 | 0.000 | 1.000 | 0.973 | 1.000 | 1.000 | 1.000 | 1.000 | 1.000 | 0.806 | 0.993 | 0.000 | 1.000 |
| | SilentCipher | 0.000 | 0.596 | 0.863 | 0.807 | 0.460 | 0.979 | 0.000 | 0.000 | 0.000 | 0.979 | 0.000 | 0.820 | 0.988 | 0.984 | 0.785 | 0.998 | 0.000 | 0.000 | 0.000 | 0.998 |
| Fish-Speech | KGW | 0.000 | 0.000 | 0.000 | 0.000 | 0.000 | 0.073 | 0.000 | 0.000 | 0.000 | 0.076 | 0.000 | 0.000 | 0.000 | 0.002 | 0.000 | 0.283 | 0.000 | 0.000 | 0.000 | 0.283 |
| | Unigram | 0.196 | 0.060 | 0.027 | 0.143 | 0.000 | 0.955 | 0.016 | 0.074 | 0.002 | 0.958 | 0.345 | 0.096 | 0.188 | 0.500 | 0.011 | 0.997 | 0.038 | 0.085 | 0.000 | 0.996 |
| | SWEET | 0.000 | 0.000 | 0.000 | 0.000 | 0.000 | 0.068 | 0.000 | 0.000 | 0.000 | 0.066 | 0.000 | 0.000 | 0.000 | 0.002 | 0.000 | 0.273 | 0.000 | 0.000 | 0.000 | 0.287 |
| | MorphMark | 0.000 | 0.000 | 0.000 | 0.000 | 0.000 | 0.000 | 0.000 | 0.000 | 0.000 | 0.000 | 0.000 | 0.000 | 0.000 | 0.000 | 0.000 | 0.000 | 0.000 | 0.000 | 0.000 | 0.000 |
| | SynthID | 0.001 | 0.004 | 0.007 | 0.016 | 0.001 | 0.441 | 0.004 | 0.001 | 0.000 | 0.460 | 0.000 | 0.014 | 0.035 | 0.104 | 0.001 | 0.838 | 0.012 | 0.002 | 0.000 | 0.840 |
| | EXP | 0.000 | 0.087 | 0.052 | 0.176 | 0.002 | 0.953 | 0.020 | 0.000 | 0.000 | 0.961 | 0.000 | 0.065 | 0.267 | 0.458 | 0.004 | 0.993 | 0.027 | 0.000 | 0.000 | 0.992 |
| | WavMark | 0.936 | 0.994 | 0.765 | 0.955 | 0.982 | 1.000 | 0.000 | 0.105 | 0.000 | 1.000 | 0.980 | 1.000 | 0.956 | 0.999 | 1.000 | 1.000 | 0.000 | 0.205 | 0.000 | 1.000 |
| | Timbre | 1.000 | 1.000 | 1.000 | 1.000 | 0.994 | 1.000 | 0.483 | 0.999 | 0.000 | 1.000 | 1.000 | 1.000 | 1.000 | 1.000 | 1.000 | 1.000 | 0.596 | 1.000 | 0.000 | 1.000 |
| | AudioSeal | 0.993 | 1.000 | 1.000 | 1.000 | 0.997 | 1.000 | 1.000 | 0.997 | 0.000 | 1.000 | 0.973 | 1.000 | 1.000 | 1.000 | 0.999 | 1.000 | 1.000 | 0.998 | 0.000 | 1.000 |
| | SilentCipher | 0.000 | 0.763 | 0.905 | 0.842 | 0.397 | 0.965 | 0.000 | 0.001 | 0.000 | 0.967 | 0.000 | 0.763 | 0.905 | 0.842 | 0.397 | 0.965 | 0.000 | 0.001 | 0.000 | 0.967 |
| Spark-TTS | KGW | 0.040 | 0.005 | 0.212 | 0.316 | 0.045 | 0.436 | 0.024 | 0.007 | 0.000 | 0.449 | 0.047 | 0.001 | 0.257 | 0.362 | 0.031 | 0.472 | 0.013 | 0.006 | 0.000 | 0.482 |
| | Unigram | 0.222 | 0.048 | 0.701 | 0.800 | 0.073 | 0.842 | 0.353 | 0.077 | 1.000 | 0.851 | 0.246 | 0.035 | 0.889 | 0.945 | 0.031 | 0.964 | 0.591 | 0.095 | 0.999 | 0.964 |
| | SWEET | 0.169 | 0.013 | 0.203 | 0.329 | 0.034 | 0.457 | 0.020 | 0.014 | 0.014 | 0.465 | 0.320 | 0.007 | 0.292 | 0.400 | 0.018 | 0.538 | 0.048 | 0.007 | 0.005 | 0.535 |
| | MorphMark | 0.034 | 0.006 | 0.028 | 0.026 | 0.032 | 0.032 | 0.029 | 0.006 | 0.000 | 0.032 | 0.056 | 0.002 | 0.019 | 0.037 | 0.012 | 0.048 | 0.006 | 0.004 | 0.000 | 0.044 |
| | SynthID | 0.017 | 0.003 | 0.147 | 0.214 | 0.004 | 0.348 | 0.034 | 0.008 | 1.000 | 0.365 | 0.000 | 0.000 | 0.000 | 0.000 | 0.001 | 0.001 | 0.000 | 0.001 | 0.877 | 0.000 |
| | EXP | 0.023 | 0.066 | 0.528 | 0.606 | 0.088 | 0.652 | 0.260 | 0.085 | 0.003 | 0.663 | 0.016 | 0.045 | 0.832 | 0.870 | 0.077 | 0.889 | 0.447 | 0.109 | 0.000 | 0.895 |
| | WavMark | 0.179 | 0.997 | 0.993 | 0.997 | 0.948 | 1.000 | 0.000 | 0.057 | 0.000 | 0.997 | 0.321 | 0.996 | 0.995 | 0.996 | 0.992 | 1.000 | 0.000 | 0.115 | 0.000 | 0.996 |
| | Timbre | 0.999 | 1.000 | 1.000 | 1.000 | 1.000 | 1.000 | 0.405 | 0.997 | 0.000 | 1.000 | 1.000 | 1.000 | 1.000 | 1.000 | 1.000 | 1.000 | 0.225 | 0.997 | 0.000 | 1.000 |
| | AudioSeal | 0.980 | 0.994 | 1.000 | 1.000 | 0.999 | 1.000 | 0.000 | 0.902 | 0.000 | 1.000 | 0.965 | 0.996 | 1.000 | 1.000 | 1.000 | 1.000 | 0.085 | 0.950 | 0.000 | 1.000 |
| | SilentCipher | 0.000 | 0.476 | 0.502 | 0.822 | 0.336 | 0.952 | 0.000 | 0.000 | 0.000 | 0.957 | 0.000 | 0.700 | 0.847 | 0.974 | 0.592 | 0.990 | 0.000 | 0.000 | 0.000 | 0.990 |

In-processing.  Post-processing.  Red Low robustness ($< 0.3$).  Orange Medium robustness ($0.3 - 0.5$).  Green High robustness ($> 0.7$).

## 4.3 DISCUSSION

**Understanding Architecture Dependent Watermark Detectability**  To investigate the underlying factors that cause in-processing watermarks to exhibit different levels of detectability across LLM-based TTS models, we conduct a systematic analysis of the token-to-audio-to-token reconstruction process. We quantify four key aspects: (1) reconstruction accuracy by comparing original language model outputs against tokens reconstructed from synthesized audio, (2) average token length, which is determined by the average audio duration and the generated tokens per audio length, (3) average duration of generated audio samples, and (4) tokens per second in the generated audio.

As depicted in Figure 4, we compute the results based on the CV3-Eval dataset for both English and Chinese samples. Our analysis shows that the architectural properties of a model are strongly correlated with watermark detectability. FireRedTTS demonstrates optimal conditions with high reconstruction accuracy and long token sequences, while Fish-Speech shows comparable reconstruction accuracy but produces shorter tokens. In contrast, Spark-TTS performs the worst, with both low reconstruction accuracy and very short token length. To capture these effects, we define "valid tokens" as the product of reconstruction accuracy and token length, which reflects both fidelity and the quantity of preserved information through the synthesis pipeline. As a result, FireRedTTS achieves the highest valid token count, followed Fish-Speech, while Spark-TTS lags far behind. This measure serves as a strong indicator of watermark detectability across architectures of LLM-based TTS models, as higher valid token counts allow watermarks to be preserved more reliably.

**Understanding Poor Robustness of In-processing Watermarks**  The key difference between LLM-based watermarks in text generation and their adaptation to LLM-based TTS models is the additional token-to-waveform synthesis step. We evaluate token reconstruction accuracy across LLM-based TTS models under various attacks to understand this relationship. As shown in Figure 5, most attacks significantly lower the token reconstruction accuracy across all models, with only MP3 compression and low-pass filtering showing minimal impact. Thus, the token reconstruction accuracy directly correlates with attack robustness. For example, the EXP watermark on FireRedTTS maintains high detection rates under attacks that preserve tokens (MP3 compression, low-pass filtering) but fails under attacks that corrupt tokens (quantization, time stretching, high-pass filtering). This explains why the in-processing watermarks struggle in LLM-based TTS models. Unlike text watermarking, where watermarked tokens remain in their original domain, LLM-based TTS models must preserve token-level information through an additional synthesis and reconstruction pipeline. Attacks that disrupt this pipeline break the fundamental assumption of token-level watermarking,

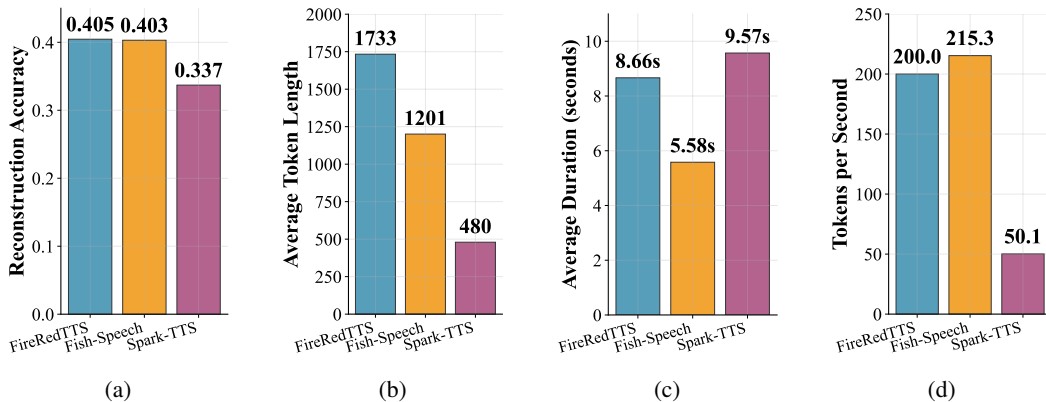

Figure 4: TTS Model Performance Comparison on CV3-Eval dataset: (a) Token reconstruction accuracy, (b) Average token length, (c) Average audio duration, (d) Token generation rate.

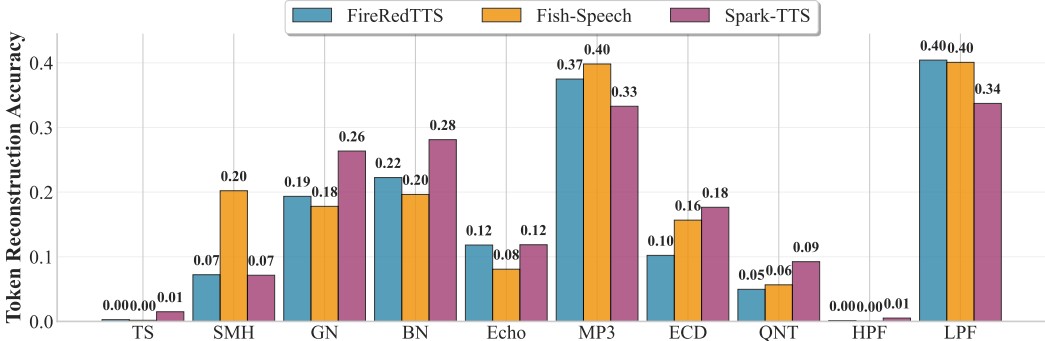

Figure 5: Token reconstruction accuracy under different attacks on CV3-Eval dataset.

leading to poor robustness of in-processing methods compared to post-processing baselines that embed watermarks directly in the speech waveform.

## 5 CONCLUSION

We present SpeechWakBench, the first benchmark evaluating in-processing watermark transferability from LLMs to speech synthesis. We compare 6 in-processing and 4 post-processing methods across 3 SOTA LLM-based TTS models under 10 attacks, introduce 16 reference-free quality metrics for unbiased evaluation, and establish a unified detectability evaluation based on TPR@X%FPR. Our results show that in-processing watermarks preserve speech quality but fail under attacks due to the irreversible token-to-waveform conversion and degradation of token reconstruction accuracy, while post-processing methods are more robust at the cost of quality. These findings demonstrate that text watermarking methods cannot be directly applied to speech, highlighting the need for novel approaches that explicitly address the challenges of the token-to-waveform conversion bottleneck.

**Limitations** In this research, we applied in-processing watermarks only to LLM with autoencoder architecture-based TTS models. Some SOTA LLM-based TTS models (Du et al., 2024) use flow-matching for token-to-waveform conversion, where waveform-to-token reconstruction is more difficult. Future work could investigate inverting the flow-matching component to embed in-processing watermarks. Due to dataset size and the number of watermarking methods, we only considered no-box attacks. White-box and black-box attacks require more evaluation time, making robustness under such attacks an interesting open problem. All watermarking methods used default hyperparameters from their implementations. The influence of hyperparameter tuning on different watermarks presents another interesting research direction.

**Reproducibility Statement** To ensure the reproducibility of our research, we have open-sourced the complete SpeechWakBench codebase at `https://anonymous.4open.science/r/SpeechWakBench-1462`. This repository includes implementations of all 10 watermarking methods, the 16 reference-free quality metrics, 10 attack scenarios, and evaluation scripts for all three LLM-based TTS models. All experiments utilize publicly available datasets (Seed-TTS-Eval and CV3-Eval) and open-source models (FireRedTTS, Fish-Speech, Spark-TTS), with detailed hyperparameters documented in the appendices and the codebase.

**Ethics Statement** This research on speech watermarking aims to advance content governance for AI-generated audio, addressing critical societal needs for combating voice cloning and voice-based fraud. Large language models were used solely for language polishing and grammar correction during paper writing.

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

# A NOMENCLATURE

| | |
|---|---|
| $\alpha$ | False positive rate threshold |
| $\delta$ | Watermark strength parameter |
| $\gamma$ | Fraction of vocabulary in green-list |
| $\hat{\mathbf{x}}$ | Recovered token sequence |
| $\hat{\mathbf{y}}$ | Potentially watermarked speech |
| $\mathbf{l}_t$ | Original logits at time step $t$ |
| $\mathbf{r}^{(t)}$ | Secret vector for EXP watermark |
| $\mathbf{x}$ | Sequence of discrete speech tokens |
| $\mathbf{x}^{(g)}$ | Global tokens |
| $\mathbf{x}^{(s)}$ | Semantic tokens |
| $\mathbf{y}$ | Waveform audio |
| $\mathcal{A}_{\text{ref}}$ | Reference audio for zero-shot synthesis |
| $\mathcal{C}$ | Control attributes including speed, pitch, and emotion |
| $\mathcal{D}$ | Neural decoder/vocoder |
| $\mathcal{E}$ | Speech encoder |
| $\mathcal{F}_{\text{score}}$ | Scoring function |
| $\mathcal{G}_{k^{(t)}}$ | Green-list tokens at time step $t$ |
| $\mathcal{K}$ | Key space |
| $\mathcal{M}_{\text{LLM}}$ | Autoregressive language model |
| $\mathcal{P}_{\text{watermark}}$ | Watermark sampling distribution |
| $\mathcal{T}$ | Input text |
| $\mathcal{V}$ | Discrete token vocabulary |
| $\mathcal{V}_g$ | Global token vocabulary |
| $\mathcal{V}_s$ | Semantic token vocabulary |
| $\mathcal{W}_{\text{logit}}$ | Logit modification watermarking function |
| $\overline{\mathcal{G}}_{k^{(t)}}$ | Non-green-list tokens at time step $t$ |
| $\tau$ | Detection threshold |
| $\tilde{\mathbf{l}}_t$ | Watermarked logits at time step $t$ |
| $\tilde{p}_t$ | Watermarked probability distribution at time step $t$ |
| $c_{\text{speed}}, c_{\text{pitch}}, c_{\text{emotion}}$ | Individual control attributes |
| $H$ | Cryptographic hash function |
| $h$ | Context window size |
| $H_0$ | Null hypothesis (no watermark) |
| $H_1$ | Alternative hypothesis (watermarked) |
| $k$ | Secret key |
| $k^{(t)}$ | Context-dependent secret key at time $t$ |
| $m$ | Number of bits in watermark message |
| $N$ | Length of waveform audio |
| $p_t$ | Original probability distribution at time step $t$ |
| $p_t[v]$ | Probability of token $v$ at time step $t$ |

| $r_v^{(t)}$ | Random value for token $v$ in EXP watermark |
| $S_T$ | Detection score |
| $T$ | Total number of tokens |
| $t$ | Time step in autoregressive generation |
| $T_g$ | Number of global tokens |
| $T_s$ | Number of semantic tokens |
| $v$ | Token in vocabulary |
| $x_t$ | Token at time step $t$ |
| $x_{<t}$ | Tokens before time step $t$ |

## B  BENCHMARK DESIGN DETAILS

### B.1  LLM-BASED TTS MODELS

**FireRedTTS (Guo et al., 2024)**  FireRedTTS presents an industry-scale TTS framework based on language modeling with three core components. The Semantic-Aware Speech Tokenizer (SAST) combines HuBERT (Hsu et al., 2021) representations discretized into 40ms tokens (16,384 code-words) with ECAPA-TDNN (Desplanques et al., 2020) utterance-level embeddings for speaker characteristics. The system employs a 30-layer autoregressive transformer (400M parameters) that processes BPE-tokenized text and speaker embeddings to generate semantic tokens. For high-fidelity synthesis, a two-stage approach first converts tokens to Mel spectrograms via flow-matching (Lipman et al., 2023) or CNN decoders, then applies BigVGAN-V2 (Lee et al., 2023) super-resolution to produce 48 kHz audio.

**Fish-Speech (Guo et al., 2024)**  Fish-Speech introduces a dual autoregressive architecture trained on 720,000 hours of multilingual data, eliminating traditional G2P dependencies through direct LLM-based feature extraction. The Dual-AR design cascades a Slow Transformer for global linguistic modeling with a Fast Transformer that refines outputs through codebook embedding processing. The Firefly-GAN vocoder employs Grouped Finite Scalar Vector Quantization (GFSQ) with depth-wise separable (Howard et al., 2017) and dilated convolutions (Yu & Koltun, 2016), achieving complete codebook utilization through systematic feature partitioning and scalar quantization. The system achieves real-time factors of 1:5 on RTX 4060 mobile and 1:15 on RTX 4090, with 150ms first-packet latency through KV-cache optimization.

**Spark-TTS (Wang et al., 2025a)**  Spark-TTS proposes a unified LLM-based architecture using BiCodec, a single-stream codec that decomposes speech while maintaining compatibility with text LLMs. BiCodec generates hybrid token streams combining semantic tokens (50 TPS) from wav2vec 2.0 (Baevski et al., 2020) features processed by ConvNeXt (Liu et al., 2022) encoders, and fixed-length global tokens encoding speaker attributes via ECAPA-TDNN (Desplanques et al., 2020) with FSQ quantization. Built on Qwen2.5-0.5B (Yang et al., 2024a), the system enables direct audio synthesis without intermediate flow-matching stages. Controllable generation spans coarse categorical labels to fine-grained numerical values through chain-of-thought inference. The accompanying VoxBox dataset provides 100,000 hours of annotated speech from 29 datasets with gender, pitch, and speed annotations.

Table 4: LLM-based TTS models with their repositories and checkpoints.

| Model | GitHub Repository | Hugging Face Checkpoint |
|---|---|---|
| FireRedTTS | FireRedTeam/FireRedTTS | FireRedTeam/FireRedTTS-1S |
| Fish-Speech | fishaudio/fish-speech | fishaudio/openaudio-s1-mini |
| Spark-TTS | SparkAudio/Spark-TTS | SparkAudio/Spark-TTS-0.5B |

## B.2 DATASETS

**Seed-TTS-Eval (Anastassiou et al., 2024)**   Seed-TTS-Eval is a bilingual test set specifically designed to assess zero-shot speech generation capabilities. It comprises English and Chinese samples drawn from public corpora, including 1,000 samples from Common Voice (Ardila et al., 2020) and 2,000 from DiDiSpeech-2 (Guo et al., 2021).

**CV3-Eval (Du et al., 2025)**   CV3-Eval is a multilingual benchmark for evaluating zero-shot speech synthesis in-the-wild scenarios, designed to address the limitations of existing evaluation benchmarks that primarily focus on clean, standard audio from sources like audiobooks (Du et al., 2025). The benchmark was released alongside CosyVoice 3 and is built on authentic in-the-wild reference speech from Common Voice (Ardila et al., 2020), FLUERS (Conneau et al., 2022), EmoBox (Ma et al., 2024), and web-crawled real-world audio data, spanning a broad range of languages and dialects, domains and environments, emotions and styles. CV3-Eval includes both objective and subjective evaluation subsets, with the objective evaluation covering three main areas: multilingual voice cloning (supporting 9 languages including Chinese, English, Japanese, Korean, German, French, Russian, Italian, and Spanish), cross-lingual voice cloning (where source audio and target text are from different languages), and emotion cloning (featuring happy, sad, and angry emotions from Chinese and English samples). This benchmark is specifically designed to evaluate the comprehensive capability of text-to-speech systems beyond traditional metrics, including aspects such as emotion expression, rhythmic richness, voice controllability, and cross-lingual voice cloning, particularly in challenging real-world scenarios with noisy backgrounds and diverse acoustic conditions (Du et al., 2025).

Table 5: Datasets with their repositories.

| Dataset | GitHub Repository |
| --- | --- |
| Seed-TTS-Eval | BytedanceSpeech/seed-tts-eval |
| CV3-Eval | FunAudioLLM/CV3-Eval |

## B.3 WATERMARKING METHODS

**KGW (Kirchenbauer et al., 2023)**   KGW watermarking establishes the foundational paradigm for LLM watermarking through vocabulary partitioning and statistical bias injection. At each generation step $t$, the vocabulary $\mathcal{V}$ is dynamically partitioned using a pseudorandom function: $\mathcal{G}_t, \mathcal{R}_t = \text{partition}(\mathcal{V}, \text{hash}(s^{(t-1)}), \gamma)$ where $\mathcal{G}_t$ represents the green list (size $\gamma|\mathcal{V}|$ with green list fraction $\gamma$), $\mathcal{R}_t$ the red list, and $s^{(t-1)}$ the previous token serving as context key. The original logits $l_k^{(t)}$ are modified through soft watermarking: $\hat{l}_k^{(t)} = l_k^{(t)} + \delta$ for tokens in $\mathcal{G}_t$, unchanged for red-list tokens, where $\delta$ is the watermark strength parameter. Detection employs a one-proportion z-test: $z = \frac{|s|_{\mathcal{G}} - \gamma T}{\sqrt{T\gamma(1-\gamma)}}$ where $|s|_{\mathcal{G}}$ represents observed green tokens in sequence $s$ and $T$ is the total token count. The method provides training-free implementation with public detectability without model access, though it suffers reduced effectiveness on low-entropy text and vulnerability to paraphrasing attacks.

**Unigram (Zhao et al., 2024)**   Unigram watermarking simplifies KGW by eliminating context dependency through fixed partitioning strategy: $\mathcal{G}, \mathcal{R} = \text{partition}(\mathcal{V}, \text{key}, \gamma)$ with no dependency on previous tokens, where key is a secret key. The detection statistic becomes more robust: $z = \frac{|s|_{\mathcal{G}} - \gamma T}{\sqrt{T\gamma(1-\gamma)}}$ with fixed lists enabling straightforward analysis and eliminating attack amplification effects present in KGW where larger context windows can amplify vulnerabilities. The method provides provable robustness against text editing attacks (insertion, deletion, substitution), paraphrasing attacks with bounded edit distance, and token-level adversarial modifications, with quality preservation mathematically proven when watermark parameter $\delta$ is appropriately chosen. Implementation benefits include better robustness against adversarial attacks, simpler theoretical analysis, no attack amplification problems, and more predictable behavior, though with trade-offs including potentially lower watermark entropy and possible vulnerability to brute-force key discovery.

**SWEET (Lee et al., 2024)** SWEET addresses the fundamental challenge of watermarking low-entropy text through entropy-based selective application. The method calculates entropy of the probability distribution at each step: $H(p^{(t)}) = -\sum_k p_k^{(t)} \log p_k^{(t)}$ where $p_k^{(t)}$ is the probability of token $k$ at step $t$, and applies watermarking only when $H(p^{(t)}) > \tau$ where $\tau$ is the entropy threshold. Detection with entropy filtering modifies the test statistic: $z = \frac{|s|_{\mathcal{G}}^{\tau} - \gamma |s|^{\tau}}{\sqrt{|s|^{\tau} \gamma (1-\gamma)}}$ where $|s|^{\tau}$ represents tokens exceeding the entropy threshold and $|s|_{\mathcal{G}}^{\tau}$ counts green tokens among high-entropy tokens. This preserves quality in structured text while maintaining detectability in high-entropy regions. Advanced development includes EWD (Entropy-based Watermark Detection) improving upon SWEET using continuous weighting functions instead of binary thresholds: $w_i = f(\text{entropy}_i)$ where $w_i$ is the weight for token $i$, with weighted detection score $z' = \frac{\sum_i w_i \cdot \mathbf{1}[\text{token}_i \in \mathcal{G}]}{\sqrt{\sum_i w_i^2}}$ where $\mathbf{1}[\cdot]$ is the indicator function.

**MorphMark (Wang et al., 2025b)** MorphMark introduces adaptive watermarking strength adjustment based on real-time entropy analysis. The method calculates cumulative green-list probability $P_{\text{green}} = \sum_{i \in \mathcal{G}} p_i$ where $p_i$ is the probability of token $i$ in the green list, and dynamically adjusts watermark strength: $\delta_t = f(P_{\text{green}})$ where $f$ is an adaptation function. The multi-objective optimization framework balances effectiveness $E$ and quality $Q$: $\max \alpha E(\delta_t) + \beta Q(\delta_t)$ subject to entropy constraints, where $\alpha$ and $\beta$ are weighting parameters. Dynamic strategy selection applies strong watermarking ($\delta_t = \delta_{\max}$) in high-entropy contexts, graduated strength ($\delta_t = \delta_{\text{base}} \times P_{\text{green}}$) in medium entropy, and reduced watermarking in low-entropy scenarios, where $\delta_{\max}$ and $\delta_{\text{base}}$ are predefined strength levels. Key distinguishing features from baseline methods include real-time adaptation versus static parameters, context awareness through cumulative probability analysis, and unified framework handling diverse entropy scenarios without preprocessing, achieving superior quality-detectability trade-off across entropy ranges with minimal computational overhead (¡0.5%).

**SynthID-Text (Dathathri et al., 2024)** SynthID-Text employs tournament sampling with pseudorandom g-functions for embedding statistical signatures during generation. The method generates random seeds: $r_t = h(x_{t-H}, ..., x_{t-1}, k)$ where $h$ is a hash function, $x_{t-H}, ..., x_{t-1}$ represents the context window of size $H$, and $k$ is the secret key, then computes g-values: $g_\ell(x, r) = F_g^{-1}\left(\frac{h(x, \ell, r)}{2^n}\right)$ for each tournament layer $\ell$, where $F_g^{-1}$ is the inverse cumulative distribution function, $x$ is a token, and $n$ is the hash output bit length. The tournament sampling algorithm samples $2^m$ tokens from LLM distribution where $m$ is the number of tournament layers, then for each layer $\ell = 1$ to $m$: groups tokens into pairs, selects winners using $g_\ell$ scores, and advances winners until the final winner becomes output token. Detection uses mean G-score: $S(x) = \frac{1}{T} \sum_{t=1}^{T} \frac{1}{m} \sum_{\ell=1}^{m} g_\ell(x_t, r_t)$ where $x_t$ is the token at position $t$, with Bayesian classification providing multi-state output: {watermarked, not watermarked, uncertain}. Production implementation features logits processor architecture integrated with Hugging Face Transformers, speculative sampling compatibility, and multi-state detection system with configurable thresholds, validated on 20 million Gemini responses with no quality degradation and formal non-distortion properties.

**EXP (Aaronson & Kirchner, 2022)** The EXP watermark utilizes exponential minimum sampling based on Gumbel noise for pseudorandom but biased token selection. For each token $x_i$, the method generates Gumbel noise $G_i \sim \text{Gumbel}(0, 1)$ and computes scores: $S_i = \log(p_i) + G_i$ where $p_i$ is the LLM probability for token $x_i$. The Gumbel-based selection process computes pseudorandom Gumbel sample $G_i = -\log(-\log(U_i))$ where $U_i = \text{PRF}(x_i, \text{context\_hash}, \text{secret\_key})$ with PRF being a pseudorandom function, then computes adjusted scores $S_i = \log(p_i) + G_i$ and selects token with maximum score: $x^* = \arg\max(S_i)$. The mathematical foundation leverages max-stable properties of Gumbel distributions for consistent sampling and exponential minimum principle providing theoretical guarantee of distribution preservation under expectation. Key advantages include strong mathematical basis in extreme value theory, provably maintains expected token distributions, minimal overhead beyond pseudorandom number generation, and straightforward implementation, though with limitations including key dependency for detection, vulnerability to synonym substitution attacks, and reduced effectiveness in low-entropy contexts.

**WavMark (Chen et al., 2023)** WavMark employs invertible neural networks (INNs) for spectrogram domain watermarking with shared parameters between encoding and decoding processes. The method transforms audio waveform $x_{\text{wave}}$ to spectrogram: $S, P = \text{STFT}(x_{\text{wave}}) \in \mathbb{R}^{2 \times T \times F}$ where $S$ and $P$ are magnitude and phase components, $T$ is time frames, and $F$ is frequency bins, and expands watermark message: $W' = \text{Linear}(W) \to \text{STFT} \to \mathbb{R}^{2 \times T \times F}$ where $W$ is the original watermark message. Invertible block operations for the $i$-th block follow: $y_1^i = x_1^i \odot \sigma(F_i(x_2^i)) + G_i(x_2^i)$ and $y_2^i = x_2^i \odot \sigma(H_i(y_1^i)) + I_i(y_1^i)$ where $\odot$ denotes element-wise multiplication, $\sigma$ is an activation function, and $F_i, G_i, H_i, I_i$ are learnable dense blocks for the $i$-th invertible block. Network architecture features 8 cascaded invertible blocks, each containing 5 layers of 2D CNNs with dense connections, window size of 1,000 samples with 400-sample hop length, and 32 bits per second capacity. Synchronization mechanism uses Brute Force Detection (BFD) combining pattern bits (10 bits) with payload bits (22 bits), achieving 0.54% BER (Bit Error Rate) localization accuracy with loss functions: $L_{\text{total}} = \lambda_1 \cdot L_{\text{message}} + \lambda_2 \cdot L_{\text{perceptual}} + \lambda_3 \cdot L_{\text{adversarial}}$ where $\lambda_1, \lambda_2, \lambda_3$ are weighting coefficients.

**Timbre (Liu et al., 2024c)** Timbre Watermarking focuses on frequency-temporal watermarking with emphasis on voice cloning attack detection. The method processes linear spectrogram as carrier: $s, p = \text{STFT}(a)$ where $a$ is the input audio, $s$ is the magnitude spectrogram, and $p$ is the phase, and performs feature extraction: $f_c = \text{EN}_c(s)$ for carrier features and $f_w = \text{EN}_w(w)$ for watermark features, where $\text{EN}_c$ and $\text{EN}_w$ are encoder networks and $w$ is the watermark message. Repeated embedding strategy follows: $f^+ = \text{Concatenate}(f_c, s, \text{Repeat}(f_w, T))$ where $T$ is the number of time frames, followed by watermark embedding: $s_w = \text{EM}(f^+)$ where EM is the embedding network and $s_w$ is the watermarked spectrogram. Extraction with averaging: $f'_w = \text{EX}(s_w)$ and $w' = \text{DE}(\text{Average}(f'_w))$ where EX is the extraction network, DE is the decoder, and $w'$ is the recovered watermark, provides temporal invariance against time-domain manipulations. Key innovation includes distortion layer simulating voice cloning pipeline during training: $\text{DP}(a_w) = \text{GL}(\text{Mel}(a_w / \max(|a_w|)))$ where DP is the distortion process, $a_w$ is watermarked audio, GL is Griffin-Lim vocoder, and Mel represents mel-spectrogram transform. The method achieves temporal invariance through repeated embedding across time frames, averaging extraction reducing time-domain sensitivity, 90% cropping robustness maintenance, 100% accuracy against professional attacks (Tacotron2, FastSpeech2 + HiFi-GAN) and 99%+ accuracy against regular attacks.

**AudioSeal (Roman et al., 2024)** AudioSeal employs generator-detector architecture trained jointly for localized watermark detection at sample level. The generator uses EnCodec-based encoder-decoder design with four convolutional blocks, residual units, LSTM layers, and ELU activation, while the decoder mirrors encoder structure using transposed convolutions and the detector outputs watermark probability at $\frac{1}{16,000}$ second resolution (corresponding to 16 kHz sampling rate). Joint optimization strategy balances perceptual loss functions minimizing difference between original and watermarked audio, and detection loss functions maximizing accuracy and localization precision. Training augmentation includes watermark masking with random selections (revert to original: 0.4, replace with zeros: 0.2, substitute different audio: 0.4) where the probabilities indicate the fraction of samples for each augmentation type, and extensive audio augmentations including bandpass filtering, echo, noise addition, and compression. The method achieves single-pass detector design with 2 orders of magnitude faster detection, sample-level resolution versus coarse 1-second alternatives, no synchronization requirements, multi-bit watermarking supporting up to 16-bit secret messages, up to $100\times$ faster detection than existing methods, sample-level watermark localization, state-of-the-art robustness against audio manipulations, and generalizability across different models and languages without retraining.

**SilentCipher (Singh et al., 2024)** SilentCipher represents the first deep learning-based model integrating psychoacoustic model-based thresholding for imperceptible watermarking. The neural network architecture includes Message Transformation Network (L) transforming message tokens $M$ into learnable embeddings $M_e(M)$ where $M_e$ is the embedding function, Encoder Network (E) processing carrier signal magnitude spectrogram combined with original carrier and message embeddings, Decoder Network reconstructing watermarked audio from combined representation, and Detector Network identifying watermark presence and extracting embedded messages. Psychoacoustic model-based thresholding uses masking threshold calculation to determine imperceptible embedding regions, band-limited signal handling addressing artifacts in frequency-limited audio,

and professional audio compatibility ensuring imperceptibility in high-quality settings. The system features pseudo-differentiable compression layers enhancing robustness against MP3, AAC, and other lossy formats while allowing gradient-based optimization despite non-differentiable operations, with SDR (Signal-to-Distortion Ratio) control mechanism providing dynamic thresholding where psychoacoustic model determines embedding capacity per segment, user-configurable SDR threshold without model retraining, frequency-aware embedding with different strengths across frequency bands, and professional quality maintenance in band-limited signals.

Our implementations of in-processing watermarking methods are adapted from the MarkLLM toolkit (Pan et al., 2024).

## B.4 WATERMARKING ATTACKS

To assess robustness, we evaluate watermarking methods under 10 attack scenarios based on the implementation from AudioMarkBench (Liu et al., 2024d). To summarize, we list the details of attacks in Table 6.

Table 6: Details of attacks.

| Attack | Parameter | Value | Description |
|---|---|---|---|
| Time stretch | Speed factor | 1.5 | Change playback speed of the audio. |
| Smooth | Window size | 6 | Apply Gaussian smoothing via 1D convolution. |
| Gaussian noise | SNR (dB) | 40 | Add random noise at a fixed SNR. |
| Background noise | SNR (dB) | 40 | Mix background noise at a fixed SNR. |
| Echo | Delay (second) | 0.9 | Introduce delayed and decayed repetitions. |
| MP3 compression | Bitrate (kbps) | 40 | Compression with the MP3 codec. |
| EnCodec | Bandwidth (kHz) | 24 | Compression with a neural audio codec. |
| Quantization | Bit levels | 64 | Reduce audio resolution to $n$ discrete levels. |
| High-pass filter | Cutoff ratio | 0.5 | Remove low frequency components. |
| Low-pass filter | Cutoff ratio | 0.5 | Remove high frequency components. |

## B.5 METRICS

We assess the quality of watermarked or non-watermarked synthetic speech by following the VERSA benchmark (Shi et al., 2025). Table 7 shows the 16 reference-free quality metrics used in our experiments.

Table 7: Details of quality metrics.

| Name | Abbreviation | Direction | Reference |
|---|---|---|---|
| Word Error Rate | WER | ↓ | Anastassiou et al. (2024) |
| Speaker Similarity | SIM | ↑ | Jung et al. (2024) |
| UTokyo-SaruLab System for VoiceMOS 2022 | UT | ↑ | Saeki et al. (2022) |
| Packet Loss Concealment-focus MOS | PLC | ↑ | Diener et al. (2023) |
| Deep Noise Suppression MOS Score of P.835 | DNS | ↑ | Reddy et al. (2022) |
| Torch-Squim PESQ | PESQ | ↑ | Kumar et al. (2023) |
| Torch-Squim MOS | MOS | ↑ | Kumar et al. (2023) |
| Torch-Squim STOI | STOI | ↑ | Kumar et al. (2023) |
| Speech Enhancement-based SAR | SAR | ↑ | Zhang et al. (2024) |
| Speech Enhancement-based SDR | SDR | ↑ | Zhang et al. (2024) |
| Speech Enhancement-based SI-SNR | SNR | ↑ | Zhang et al. (2024) |
| Torch-Squim SI-SDR | Si-SDR | ↑ | Kumar et al. (2023) |
| Subjective Speech Quality Assessment | SSQA | ↑ | Huang et al. (2024) |
| Deep Noise Suppression MOS Score of P.808 | DNS-P | ↑ | Reddy et al. (2022) |
| Singing voice MOS | SING | ↑ | Tang et al. (2024) |
| Speech Enhancement-based CI-SDR | Ci-SDR | ↑ | Zhang et al. (2024) |

[↑] Higher is better.  [↓] Lower is better.

## C  ADDITIONAL EXPERIMENTAL RESULTS

Table 8: Quality and detectability performance of watermarking methods on CV3-Eval.

| Model | Method | EN Quality Metric WER↓ | SIM↑ | UT↑ | PLC↑ | DNS↑ | PESQ↑ | TPR@X%FPR 0.2%↑ | 3.0%↑ | ZH Quality Metric WER↓ | SIM↑ | UT↑ | PLC↑ | DNS↑ | PESQ↑ | TPR@X%FPR 0.2%↑ | 3.0%↑ |
|---|---|---|---|---|---|---|---|---|---|---|---|---|---|---|---|---|---|
| FireRedTTS | No Watermark | 6.8 | 0.62 | 3.13 | 3.98 | 3.00 | 2.92 | — | — | 4.4 | 0.68 | 2.55 | 4.10 | 3.12 | 3.04 | — | — |
| | KGW | 6.6 | 0.62 | 3.14 | 3.97 | 3.01 | 2.94 | 0.980 | 0.990 | 4.5 | 0.68 | 2.55 | 4.09 | 3.12 | 3.04 | 0.994 | 0.998 |
| | Unigram | 6.4 | 0.62 | 3.12 | 3.96 | 2.99 | 2.90 | 1.000 | 1.000 | 4.6 | 0.67 | 2.54 | 4.10 | 3.12 | 3.04 | 1.000 | 1.000 |
| | SWEET | 6.4 | 0.62 | 3.11 | 3.95 | 3.00 | 2.94 | 0.988 | 0.996 | 4.7 | 0.68 | 2.55 | 4.09 | 3.12 | 3.03 | 0.998 | 1.000 |
| | MorphMark | 6.6 | 0.62 | 3.15 | 3.97 | 2.99 | 2.92 | 0.128 | 0.394 | 4.1 | 0.68 | 2.56 | 4.11 | 3.12 | 3.05 | 0.437 | 0.673 |
| | SynthID | 7.2 | 0.62 | 3.13 | 3.97 | 3.00 | 2.92 | 0.984 | 0.996 | 4.7 | 0.68 | 2.53 | 4.09 | 3.11 | 3.04 | 0.996 | 1.000 |
| | EXP | 5.5 | 0.65 | 3.61 | 4.11 | 3.10 | 3.17 | 1.000 | 1.000 | 3.9 | 0.70 | 3.08 | 4.24 | 3.20 | 3.27 | 1.000 | 1.000 |
| | WavMark | 6.8 | 0.63 | 3.07 | 3.97 | 2.98 | 2.81 | 0.990 | 0.990 | 4.5 | 0.69 | 2.50 | 4.05 | 3.10 | 2.90 | 1.000 | 1.000 |
| | Timbre | 6.9 | 0.60 | 2.92 | 3.72 | 2.88 | 2.52 | 1.000 | 1.000 | 4.4 | 0.67 | 2.44 | 3.92 | 3.02 | 2.79 | 1.000 | 1.000 |
| | AudioSeal | 7.0 | 0.62 | 3.10 | 3.99 | 2.95 | 2.89 | 1.000 | 1.000 | 4.4 | 0.68 | 2.53 | 4.11 | 3.10 | 2.99 | 1.000 | 1.000 |
| | SilentCipher | 6.9 | 0.51 | 2.86 | 3.93 | 2.88 | 2.79 | 1.000 | 1.000 | 4.5 | 0.68 | 2.55 | 4.12 | 3.12 | 3.06 | 1.000 | 1.000 |
| Fish-Speech | No Watermark | 5.3 | 0.50 | 3.87 | 4.42 | 3.15 | 3.39 | — | — | 4.0 | 0.59 | 3.22 | 4.43 | 3.22 | 3.41 | — | — |
| | KGW | 4.6 | 0.49 | 3.87 | 4.42 | 3.16 | 3.42 | 0.122 | 0.348 | 5.0 | 0.59 | 3.19 | 4.42 | 3.21 | 3.39 | 0.214 | 0.471 |
| | Unigram | 5.0 | 0.50 | 3.88 | 4.41 | 3.16 | 3.40 | 0.960 | 0.978 | 4.2 | 0.59 | 3.21 | 4.42 | 3.22 | 3.39 | 0.992 | 0.994 |
| | SWEET | 5.3 | 0.50 | 3.85 | 4.41 | 3.15 | 3.39 | 0.136 | 0.334 | 4.1 | 0.59 | 3.19 | 4.41 | 3.21 | 3.39 | 0.260 | 0.487 |
| | MorphMark | 5.1 | 0.50 | 3.87 | 4.42 | 3.16 | 3.39 | 0.000 | 0.000 | 4.1 | 0.59 | 3.22 | 4.43 | 3.21 | 3.41 | 0.000 | 0.000 |
| | SynthID | 4.9 | 0.50 | 3.87 | 4.41 | 3.16 | 3.41 | 0.584 | 0.844 | 4.1 | 0.59 | 3.18 | 4.42 | 3.20 | 3.40 | 0.788 | 0.926 |
| | EXP | 4.4 | 0.52 | 4.02 | 4.45 | 3.20 | 3.53 | 0.890 | 0.944 | 3.7 | 0.61 | 3.42 | 4.45 | 3.25 | 3.51 | 0.978 | 0.986 |
| | WavMark | 5.4 | 0.51 | 3.76 | 4.34 | 3.09 | 3.17 | 1.000 | 1.000 | 4.0 | 0.60 | 3.11 | 4.32 | 3.13 | 3.14 | 1.000 | 1.000 |
| | Timbre | 5.6 | 0.47 | 3.69 | 4.20 | 3.04 | 3.10 | 1.000 | 1.000 | 4.0 | 0.57 | 3.08 | 4.21 | 3.15 | 3.22 | 1.000 | 1.000 |
| | AudioSeal | 5.5 | 0.50 | 3.84 | 4.40 | 3.09 | 3.33 | 1.000 | 1.000 | 3.9 | 0.59 | 3.19 | 4.42 | 3.18 | 3.35 | 1.000 | 1.000 |
| | SilentCipher | 5.5 | 0.51 | 3.87 | 4.44 | 3.15 | 3.42 | 0.966 | 0.966 | 4.0 | 0.59 | 3.22 | 4.45 | 3.23 | 3.43 | 0.998 | 0.998 |
| Spark-TTS | No Watermark | 9.8 | 0.55 | 3.46 | 3.75 | 2.94 | 3.00 | — | — | 5.9 | 0.68 | 2.96 | 4.00 | 3.15 | 3.17 | — | — |
| | KGW | 9.0 | 0.57 | 3.54 | 3.81 | 3.03 | 3.07 | 0.434 | 0.630 | 5.1 | 0.68 | 2.92 | 4.01 | 3.15 | 3.17 | 0.528 | 0.776 |
| | Unigram | 13.9 | 0.55 | 3.49 | 3.69 | 2.94 | 3.03 | 0.816 | 0.896 | 9.0 | 0.66 | 2.91 | 3.92 | 3.10 | 3.09 | 0.956 | 0.978 |
| | SWEET | 11.1 | 0.56 | 3.51 | 3.75 | 2.99 | 3.05 | 0.400 | 0.594 | 5.6 | 0.68 | 2.93 | 3.99 | 3.15 | 3.14 | 0.574 | 0.766 |
| | MorphMark | 10.4 | 0.56 | 3.54 | 3.79 | 2.99 | 3.07 | 0.076 | 0.226 | 5.1 | 0.68 | 2.96 | 4.01 | 3.15 | 3.16 | 0.060 | 0.208 |
| | SynthID | 8.9 | 0.57 | 3.50 | 3.77 | 3.01 | 3.07 | 0.006 | 0.508 | 4.8 | 0.69 | 2.92 | 4.00 | 3.15 | 3.15 | 0.006 | 0.634 |
| | EXP | 33.8 | 0.38 | 3.11 | 3.15 | 2.43 | 2.56 | 0.546 | 0.606 | 31.7 | 0.46 | 2.65 | 3.41 | 2.61 | 2.74 | 0.668 | 0.720 |
| | WavMark | 9.8 | 0.58 | 3.50 | 3.81 | 2.99 | 2.97 | 0.986 | 0.986 | 5.7 | 0.69 | 2.87 | 3.96 | 3.13 | 3.05 | 0.996 | 0.996 |
| | Timbre | 9.8 | 0.57 | 3.42 | 3.59 | 2.86 | 2.77 | 1.000 | 1.000 | 6.1 | 0.68 | 2.81 | 3.69 | 3.00 | 2.92 | 1.000 | 1.000 |
| | AudioSeal | 10.0 | 0.57 | 3.56 | 3.79 | 2.99 | 3.01 | 1.000 | 1.000 | 8.4 | 0.68 | 2.94 | 3.96 | 3.13 | 3.11 | 1.000 | 1.000 |
| | SilentCipher | 9.8 | 0.53 | 3.56 | 4.06 | 3.01 | 3.10 | 0.932 | 0.932 | 7.7 | 0.62 | 2.97 | 4.16 | 3.13 | 3.17 | 0.984 | 0.984 |

☐ In-processing.  ☐ Post-processing.  ↑ Higher is better.  ↓ Lower is better.

Table 9: Quality comparison across watermarking methods on Seed-TTS-Eval English.

| Model | Method | WER↓ | SIM↑ | UT↑ | PLC↑ | DNS↑ | MOS↑ | STOI↑ | PESQ↑ | SAR↑ | SDR↑ | SNR↑ | Si-SDR↑ | SSQA↑ | DNS-P↑ | SING↑ | Ci-SDR↑ |
|---|---|---|---|---|---|---|---|---|---|---|---|---|---|---|---|---|---|
| FireRedTTS | No Watermark | 2.5 | 0.63 | 3.64 | 4.12 | 3.04 | 4.34 | 0.98 | 3.20 | 39.06 | 39.06 | 38.45 | 20.54 | 4.21 | 3.80 | 3.69 | 38.75 |
| | KGW | 2.6 | 0.63 | 3.62 | 4.10 | 3.04 | 4.35 | 0.98 | 3.20 | 38.83 | 38.83 | 38.23 | 20.51 | 4.20 | 3.79 | 3.69 | 38.57 |
| | Unigram | 2.8 | 0.63 | 3.62 | 4.12 | 3.05 | 4.35 | 0.98 | 3.18 | 38.81 | 38.81 | 38.21 | 20.37 | 4.20 | 3.80 | 3.69 | 38.55 |
| | SWEET | 2.6 | 0.63 | 3.62 | 4.12 | 3.04 | 4.36 | 0.98 | 3.18 | 38.79 | 38.79 | 38.20 | 20.36 | 4.21 | 3.80 | 3.69 | 38.58 |
| | MorphMark | 2.5 | 0.63 | 3.63 | 4.13 | 3.04 | 4.35 | 0.98 | 3.18 | 38.85 | 38.85 | 38.25 | 20.45 | 4.21 | 3.80 | 3.69 | 38.57 |
| | SynthID | 2.6 | 0.63 | 3.61 | 4.11 | 3.04 | 4.35 | 0.98 | 3.19 | 38.82 | 38.82 | 38.21 | 20.53 | 4.20 | 3.79 | 3.69 | 38.63 |
| | EXP | 2.2 | 0.65 | 3.97 | 4.24 | 3.12 | 4.34 | 0.99 | 3.42 | 39.73 | 39.73 | 39.04 | 22.26 | 4.37 | 3.85 | 3.69 | 39.50 |
| | WavMark | 2.6 | 0.63 | 3.57 | 4.12 | 3.00 | 4.33 | 0.98 | 3.11 | 38.78 | 38.78 | 38.33 | 20.72 | 4.05 | 3.75 | 3.70 | 38.71 |
| | Timbre | 2.6 | 0.62 | 3.47 | 3.82 | 2.88 | 4.34 | 0.97 | 2.84 | 37.68 | 37.68 | 37.46 | 19.34 | 3.94 | 3.66 | 3.66 | 37.56 |
| | AudioSeal | 2.6 | 0.63 | 3.61 | 4.14 | 3.02 | 4.34 | 0.98 | 3.18 | 38.65 | 38.65 | 38.35 | 20.18 | 4.19 | 3.76 | 3.68 | 38.45 |
| | SilentCipher | 2.5 | 0.63 | 3.63 | 4.26 | 3.04 | 4.34 | 0.98 | 3.22 | 39.04 | 39.04 | 38.38 | 20.67 | 4.22 | 3.80 | 3.70 | 38.46 |
| Fish-Speech | No Watermark | 2.0 | 0.53 | 4.15 | 4.45 | 3.22 | 4.30 | 0.99 | 3.58 | 40.27 | 40.27 | 39.26 | 23.72 | 4.45 | 3.96 | 3.77 | 39.22 |
| | KGW | 1.8 | 0.53 | 4.14 | 4.44 | 3.22 | 4.31 | 0.99 | 3.58 | 40.23 | 40.23 | 39.23 | 23.70 | 4.44 | 3.96 | 3.78 | 39.22 |
| | Unigram | 2.3 | 0.53 | 4.14 | 4.44 | 3.22 | 4.30 | 0.99 | 3.58 | 40.43 | 40.43 | 39.41 | 23.64 | 4.44 | 3.96 | 3.78 | 39.46 |
| | SWEET | 2.0 | 0.53 | 4.14 | 4.44 | 3.22 | 4.31 | 0.99 | 3.59 | 40.25 | 40.25 | 39.25 | 23.78 | 4.45 | 3.95 | 3.78 | 39.10 |
| | MorphMark | 2.0 | 0.53 | 4.15 | 4.44 | 3.22 | 4.31 | 0.99 | 3.59 | 40.41 | 40.41 | 39.42 | 23.79 | 4.44 | 3.96 | 3.78 | 39.45 |
| | SynthID | 2.0 | 0.53 | 4.13 | 4.44 | 3.22 | 4.29 | 0.99 | 3.57 | 40.22 | 40.22 | 39.28 | 23.66 | 4.44 | 3.95 | 3.77 | 39.31 |
| | EXP | 2.1 | 0.54 | 4.21 | 4.46 | 3.24 | 4.30 | 0.99 | 3.66 | 40.18 | 40.18 | 38.98 | 24.12 | 4.47 | 3.97 | 3.77 | 39.05 |
| | WavMark | 2.1 | 0.55 | 4.04 | 4.46 | 3.16 | 4.27 | 0.99 | 3.38 | 40.21 | 40.21 | 39.69 | 23.51 | 4.29 | 3.87 | 3.77 | 39.62 |
| | Timbre | 2.1 | 0.51 | 4.04 | 4.32 | 3.14 | 4.32 | 0.98 | 3.39 | 40.10 | 40.10 | 39.38 | 22.83 | 4.32 | 3.84 | 3.77 | 39.31 |
| | AudioSeal | 2.1 | 0.53 | 4.14 | 4.45 | 3.19 | 4.29 | 0.99 | 3.55 | 39.81 | 39.81 | 38.84 | 23.05 | 4.43 | 3.93 | 3.76 | 38.86 |
| | SilentCipher | 2.1 | 0.54 | 4.15 | 4.45 | 3.22 | 4.31 | 0.99 | 3.60 | 40.37 | 40.37 | 39.33 | 24.07 | 4.45 | 3.96 | 3.78 | 39.37 |
| Spark-TTS | No Watermark | 2.7 | 0.59 | 3.93 | 4.39 | 3.12 | 4.31 | 0.98 | 3.31 | 40.09 | 40.09 | 39.33 | 21.81 | 4.37 | 3.78 | 3.89 | 39.89 |
| | KGW | 3.0 | 0.59 | 3.89 | 4.37 | 3.12 | 4.32 | 0.98 | 3.27 | 40.22 | 40.22 | 39.48 | 21.55 | 4.35 | 3.78 | 3.87 | 39.99 |
| | Unigram | 5.2 | 0.58 | 3.88 | 4.32 | 3.08 | 4.27 | 0.97 | 3.24 | 39.33 | 39.33 | 38.43 | 21.01 | 4.28 | 3.74 | 3.84 | 38.90 |
| | SWEET | 4.3 | 0.58 | 3.90 | 4.35 | 3.11 | 4.30 | 0.97 | 3.26 | 39.84 | 39.84 | 39.05 | 21.23 | 4.32 | 3.77 | 3.86 | 39.63 |
| | MorphMark | 2.6 | 0.59 | 3.94 | 4.39 | 3.14 | 4.30 | 0.98 | 3.31 | 40.33 | 40.33 | 39.58 | 22.00 | 4.38 | 3.79 | 3.89 | 40.00 |
| | SynthID | 2.8 | 0.60 | 3.90 | 4.38 | 3.13 | 4.30 | 0.98 | 3.30 | 40.29 | 40.29 | 39.56 | 21.73 | 4.35 | 3.77 | 3.89 | 40.08 |
| | EXP | 18.3 | 0.48 | 3.73 | 3.99 | 2.76 | 4.19 | 0.92 | 2.96 | 31.68 | 31.68 | 29.28 | 15.61 | 3.82 | 3.48 | 3.60 | 31.52 |
| | WavMark | 3.9 | 0.60 | 3.76 | 4.30 | 3.09 | 4.26 | 0.98 | 3.24 | 37.68 | 37.68 | 36.96 | 21.82 | 4.11 | 3.71 | 3.86 | 37.54 |
| | Timbre | 2.8 | 0.59 | 3.78 | 4.19 | 2.96 | 4.31 | 0.97 | 2.95 | 38.24 | 38.24 | 37.73 | 20.29 | 4.18 | 3.65 | 3.87 | 38.14 |
| | AudioSeal | 2.8 | 0.59 | 3.91 | 4.41 | 3.11 | 4.31 | 0.98 | 3.25 | 39.83 | 39.83 | 39.14 | 20.68 | 4.35 | 3.76 | 3.86 | 39.55 |
| | SilentCipher | 2.8 | 0.57 | 3.93 | 4.40 | 3.11 | 4.32 | 0.98 | 3.33 | 40.30 | 40.30 | 39.34 | 21.90 | 4.38 | 3.78 | 3.90 | 39.82 |

☐ In-processing.  ☐ Post-processing.  ↑ Higher is better.  ↓ Lower is better.

Table 10: Quality comparison across watermarking methods on Seed-TTS-Eval Chinese.

| Model | Method | WER↓ | SIM↑ | UT↑ | PLC↑ | DNS↑ | MOS↑ | STOI↑ | PESQ↑ | SAR↑ | SDR↑ | SNR↑ | Si-SDR↑ | SSQA↑ | DNS-P↑ | SING↑ | Ci-SDR↑ |
|---|---|---|---|---|---|---|---|---|---|---|---|---|---|---|---|---|---|
| FireRedTTS | No Watermark | 1.2 | 0.74 | 2.93 | 4.17 | 3.16 | 4.26 | 0.99 | 3.47 | 38.04 | 38.04 | ∞ | 23.27 | 4.29 | 3.84 | 3.85 | 38.03 |
| | KGW | 1.1 | 0.75 | 2.92 | 4.16 | 3.15 | 4.25 | 0.99 | 3.48 | 37.84 | 37.84 | ∞ | 23.28 | 4.28 | 3.84 | 3.85 | 37.84 |
| | Unigram | 1.2 | 0.74 | 2.91 | 4.16 | 3.15 | 4.26 | 0.99 | 3.47 | 37.97 | 37.97 | ∞ | 23.19 | 4.28 | 3.84 | 3.84 | 37.97 |
| | SWEET | 1.2 | 0.75 | 2.92 | 4.16 | 3.15 | 4.26 | 0.99 | 3.48 | 37.75 | 37.75 | 37.51 | 23.13 | 4.29 | 3.84 | 3.84 | 37.74 |
| | MorphMark | 1.2 | 0.74 | 2.93 | 4.17 | 3.16 | 4.25 | 0.99 | 3.47 | 38.16 | 38.16 | ∞ | 23.20 | 4.29 | 3.84 | 3.85 | 38.16 |
| | SynthID | 1.2 | 0.74 | 2.92 | 4.15 | 3.15 | 4.25 | 0.99 | 3.47 | 38.00 | 38.00 | 37.77 | 23.22 | 4.28 | 3.84 | 3.85 | 37.97 |
| | EXP | 1.0 | 0.75 | 3.29 | 4.31 | 3.22 | 4.25 | 0.99 | 3.66 | 38.85 | 38.85 | ∞ | 24.40 | 4.42 | 3.88 | 3.89 | 38.85 |
| | WavMark | 1.1 | 0.73 | 2.89 | 4.25 | 3.13 | 4.25 | 0.99 | 3.36 | 37.55 | 37.55 | ∞ | 23.57 | 4.04 | 3.82 | 3.88 | 37.55 |
| | Timbre | 1.2 | 0.73 | 2.74 | 3.81 | 2.99 | 4.25 | 0.97 | 2.98 | 35.49 | 35.49 | ∞ | 21.42 | 4.02 | 3.67 | 3.77 | 35.49 |
| | AudioSeal | 1.2 | 0.74 | 2.89 | 4.15 | 3.13 | 4.27 | 0.99 | 3.45 | 37.45 | 37.45 | ∞ | 23.29 | 4.26 | 3.79 | 3.85 | 37.45 |
| | SilentCipher | 1.2 | 0.74 | 2.94 | 4.32 | 3.16 | 4.26 | 0.99 | 3.49 | 37.59 | 37.59 | ∞ | 23.13 | 4.30 | 3.86 | 3.85 | 37.23 |
| Fish-Speech | No Watermark | 1.1 | 0.69 | 3.50 | 4.47 | 3.26 | 4.24 | 0.98 | 3.56 | 40.27 | 40.27 | 39.91 | 23.39 | 4.49 | 3.90 | 3.99 | 39.59 |
| | KGW | 1.2 | 0.68 | 3.45 | 4.47 | 3.25 | 4.25 | 0.98 | 3.56 | 40.13 | 40.13 | 39.78 | 23.26 | 4.47 | 3.90 | 3.98 | 39.62 |
| | Unigram | 1.2 | 0.69 | 3.46 | 4.46 | 3.25 | 4.25 | 0.99 | 3.55 | 40.19 | 40.19 | 39.82 | 23.34 | 4.47 | 3.90 | 3.99 | 39.57 |
| | SWEET | 1.3 | 0.68 | 3.45 | 4.46 | 3.25 | 4.24 | 0.99 | 3.56 | 40.28 | 40.28 | 39.94 | 23.41 | 4.47 | 3.90 | 3.98 | 39.71 |
| | MorphMark | 1.3 | 0.69 | 3.48 | 4.47 | 3.26 | 4.25 | 0.99 | 3.57 | 40.16 | 40.16 | 39.82 | 23.37 | 4.48 | 3.90 | 3.99 | 39.58 |
| | SynthID | 1.2 | 0.68 | 3.45 | 4.46 | 3.25 | 4.25 | 0.98 | 3.56 | 40.27 | 40.27 | 39.94 | 23.45 | 4.47 | 3.90 | 3.98 | 39.71 |
| | EXP | 1.2 | 0.69 | 3.54 | 4.49 | 3.28 | 4.24 | 0.99 | 3.61 | 40.23 | 40.23 | 39.74 | 23.56 | 4.50 | 3.91 | 3.98 | 39.62 |
| | WavMark | 1.2 | 0.69 | 3.39 | 4.52 | 3.18 | 4.22 | 0.98 | 3.35 | 39.25 | 39.25 | 39.04 | 23.80 | 4.30 | 3.84 | 3.96 | 38.82 |
| | Timbre | 1.2 | 0.67 | 3.32 | 4.30 | 3.18 | 4.25 | 0.98 | 3.25 | 39.46 | 39.46 | 39.34 | 21.97 | 4.36 | 3.73 | 3.93 | 39.01 |
| | AudioSeal | 1.2 | 0.69 | 3.46 | 4.46 | 3.21 | 4.25 | 0.98 | 3.53 | 39.55 | 39.55 | 39.20 | 23.41 | 4.47 | 3.85 | 3.97 | 39.01 |
| | SilentCipher | 1.2 | 0.69 | 3.50 | 4.47 | 3.27 | 4.24 | 0.98 | 3.57 | 40.42 | 40.42 | 40.03 | 23.55 | 4.49 | 3.92 | 3.98 | 39.79 |
| Spark-TTS | No Watermark | 1.6 | 0.67 | 3.28 | 4.38 | 3.22 | 4.04 | 0.98 | 3.58 | 39.42 | 38.78 | ∞ | 24.41 | 4.40 | 3.80 | 3.97 | 39.37 |
| | KGW | 1.5 | 0.67 | 3.24 | 4.37 | 3.22 | 4.17 | 0.99 | 3.58 | 39.39 | 39.39 | 38.95 | 24.32 | 4.41 | 3.80 | 3.97 | 39.27 |
| | Unigram | 3.6 | 0.65 | 3.36 | 4.31 | 3.18 | 4.15 | 0.98 | 3.57 | 38.39 | 38.39 | 37.83 | 24.55 | 4.35 | 3.72 | 3.96 | 38.35 |
| | SWEET | 2.7 | 0.67 | 3.23 | 4.36 | 3.21 | 4.15 | 0.98 | 3.56 | 39.48 | 39.48 | 39.07 | 24.16 | 4.39 | 3.79 | 3.96 | 39.35 |
| | MorphMark | 2.7 | 0.67 | 3.27 | 4.37 | 3.22 | 4.16 | 0.98 | 3.59 | 39.25 | 39.25 | 38.76 | 24.37 | 4.40 | 3.80 | 3.97 | 39.08 |
| | SynthID | 1.4 | 0.67 | 3.23 | 4.36 | 3.22 | 4.16 | 0.99 | 3.57 | 39.45 | 39.45 | 39.07 | 24.35 | 4.40 | 3.80 | 3.97 | 39.32 |
| | EXP | 14.0 | 0.60 | 3.14 | 4.17 | 3.02 | 4.09 | 0.96 | 3.39 | 35.38 | 35.38 | ∞ | 20.95 | 4.11 | 3.64 | 3.79 | 35.22 |
| | WavMark | 1.6 | 0.67 | 3.15 | 4.32 | 3.12 | 4.14 | 0.98 | 3.39 | 36.86 | 36.86 | 36.47 | 24.17 | 4.09 | 3.62 | 3.97 | 36.86 |
| | Timbre | 2.3 | 0.65 | 3.09 | 4.17 | 3.02 | 4.19 | 0.98 | 3.13 | 36.82 | 36.82 | 36.53 | 22.49 | 4.25 | 3.66 | 3.92 | 36.76 |
| | AudioSeal | 2.3 | 0.67 | 3.25 | 4.36 | 3.20 | 4.17 | 0.98 | 3.54 | 39.11 | 39.11 | 38.74 | 23.80 | 4.39 | 3.77 | 3.95 | 39.02 |
| | SilentCipher | 1.4 | 0.67 | 3.23 | 4.36 | 3.22 | 4.18 | 0.98 | 3.57 | 39.55 | 39.55 | 38.92 | 24.30 | 4.42 | 3.80 | 3.97 | 39.23 |

In-processing.  Post-processing.  ↑ Higher is better.  ↓ Lower is better.  ∞ Infinity.

Table 11: Quality comparison across watermarking methods on CV3-Eval English.

| Model | Method | WER↓ | SIM↑ | UT↑ | PLC↑ | DNS↑ | MOS↑ | STOI↑ | PESQ↑ | SAR↑ | SDR↑ | SNR↑ | Si-SDR↑ | SSQA↑ | DNS-P↑ | SING↑ | Ci-SDR↑ |
|---|---|---|---|---|---|---|---|---|---|---|---|---|---|---|---|---|---|
| FireRedTTS | No Watermark | 6.8 | 0.62 | 3.13 | 3.98 | 3.00 | 4.26 | 0.97 | 2.92 | 32.83 | 32.83 | 32.91 | 19.57 | 3.68 | 3.66 | 3.65 | 32.68 |
| | KGW | 6.6 | 0.62 | 3.14 | 3.97 | 3.01 | 4.26 | 0.97 | 2.94 | 33.48 | 33.48 | 33.59 | 19.94 | 3.69 | 3.66 | 3.67 | 33.33 |
| | Unigram | 6.4 | 0.62 | 3.12 | 3.96 | 2.99 | 4.27 | 0.97 | 2.90 | 33.26 | 33.26 | 33.32 | 19.47 | 3.68 | 3.64 | 3.67 | 33.16 |
| | SWEET | 6.4 | 0.62 | 3.11 | 3.95 | 3.00 | 4.27 | 0.97 | 2.94 | 33.26 | 33.26 | ∞ | 19.68 | 3.67 | 3.66 | 3.66 | 33.10 |
| | MorphMark | 6.6 | 0.62 | 3.15 | 3.97 | 2.99 | 4.26 | 0.97 | 2.92 | 33.53 | 33.53 | 33.63 | 19.79 | 3.69 | 3.65 | 3.66 | 33.33 |
| | SynthID | 7.2 | 0.62 | 3.13 | 3.97 | 3.00 | 4.26 | 0.97 | 2.90 | 33.51 | 33.51 | 33.56 | 19.50 | 3.68 | 3.64 | 3.65 | 33.34 |
| | EXP | 5.5 | 0.65 | 3.61 | 4.11 | 3.10 | 4.23 | 0.98 | 3.17 | 34.54 | 34.54 | ∞ | 21.99 | 3.90 | 3.70 | 3.65 | 34.47 |
| | WavMark | 6.8 | 0.63 | 3.07 | 3.97 | 2.98 | 4.26 | 0.96 | 2.81 | 32.24 | 32.24 | 32.40 | 19.28 | 3.51 | 3.63 | 3.70 | 32.03 |
| | Timbre | 6.9 | 0.60 | 2.92 | 3.72 | 2.88 | 4.21 | 0.95 | 2.52 | 31.10 | 31.10 | 31.13 | 17.80 | 3.38 | 3.48 | 3.68 | 30.99 |
| | AudioSeal | 7.0 | 0.62 | 3.10 | 3.99 | 2.95 | 4.25 | 0.97 | 2.89 | 32.20 | 32.20 | ∞ | 19.34 | 3.60 | 3.60 | 3.60 | 31.98 |
| | SilentCipher | 6.9 | 0.51 | 2.86 | 3.93 | 2.88 | 4.23 | 0.95 | 2.79 | 33.09 | 33.09 | 32.96 | 18.49 | 3.63 | 3.63 | 3.59 | 32.92 |
| Fish-Speech | No Watermark | 5.3 | 0.50 | 3.87 | 4.42 | 3.15 | 4.08 | 0.98 | 3.39 | 38.38 | 38.38 | 37.82 | 23.20 | 4.16 | 3.81 | 4.00 | 38.17 |
| | KGW | 4.6 | 0.49 | 3.87 | 4.42 | 3.16 | 4.07 | 0.98 | 3.42 | 38.62 | 38.62 | 38.07 | 23.27 | 4.18 | 3.81 | 4.00 | 38.49 |
| | Unigram | 5.0 | 0.50 | 3.88 | 4.41 | 3.16 | 4.06 | 0.98 | 3.40 | 38.61 | 38.61 | 38.11 | 23.21 | 4.17 | 3.81 | 3.99 | 38.40 |
| | SWEET | 5.3 | 0.50 | 3.85 | 4.41 | 3.15 | 4.07 | 0.98 | 3.39 | 38.96 | 38.96 | 38.41 | 22.95 | 4.16 | 3.81 | 3.99 | 38.70 |
| | MorphMark | 5.1 | 0.50 | 3.87 | 4.42 | 3.16 | 4.12 | 0.98 | 3.39 | 38.71 | 38.71 | 38.18 | 23.17 | 4.17 | 3.81 | 3.99 | 38.62 |
| | SynthID | 4.9 | 0.50 | 3.87 | 4.41 | 3.16 | 4.08 | 0.98 | 3.41 | 38.70 | 38.70 | 38.18 | 23.24 | 4.16 | 3.81 | 3.99 | 38.45 |
| | EXP | 4.4 | 0.52 | 4.02 | 4.45 | 3.20 | 4.08 | 0.98 | 3.53 | 38.71 | 38.71 | 38.11 | 24.03 | 4.23 | 3.80 | 4.00 | 38.25 |
| | WavMark | 5.4 | 0.51 | 3.76 | 4.34 | 3.09 | 4.06 | 0.98 | 3.17 | 37.50 | 37.50 | 37.07 | 22.82 | 3.95 | 3.75 | 3.96 | 37.22 |
| | Timbre | 5.6 | 0.47 | 3.69 | 4.20 | 3.04 | 4.14 | 0.98 | 3.10 | 37.64 | 37.64 | 37.20 | 22.06 | 3.95 | 3.65 | 3.95 | 37.46 |
| | AudioSeal | 5.5 | 0.50 | 3.84 | 4.40 | 3.09 | 4.08 | 0.98 | 3.33 | 37.46 | 37.46 | 36.91 | 22.84 | 4.13 | 3.75 | 3.99 | 37.24 |
| | SilentCipher | 5.5 | 0.51 | 3.87 | 4.44 | 3.15 | 4.09 | 0.98 | 3.42 | 38.48 | 38.48 | 37.90 | 23.43 | 4.17 | 3.82 | 4.00 | 38.27 |
| Spark-TTS | No Watermark | 9.8 | 0.55 | 3.46 | 3.75 | 2.94 | 4.07 | 0.95 | 3.00 | 36.01 | 36.01 | ∞ | 19.74 | 3.93 | 3.56 | 3.65 | 35.93 |
| | KGW | 9.0 | 0.57 | 3.54 | 3.81 | 3.03 | 4.07 | 0.96 | 3.07 | 36.32 | 36.32 | ∞ | 20.48 | 4.00 | 3.62 | 3.70 | 36.15 |
| | Unigram | 13.9 | 0.55 | 3.49 | 3.69 | 2.94 | 4.07 | 0.95 | 3.03 | 34.91 | 34.91 | ∞ | 19.23 | 3.82 | 3.54 | 3.59 | 34.75 |
| | SWEET | 11.1 | 0.56 | 3.51 | 3.75 | 2.99 | 4.10 | 0.96 | 3.05 | 35.92 | 35.92 | ∞ | 19.82 | 3.92 | 3.60 | 3.65 | 35.86 |
| | MorphMark | 10.4 | 0.56 | 3.54 | 3.79 | 2.99 | 4.09 | 0.96 | 3.07 | 35.82 | 35.82 | ∞ | 20.29 | 3.94 | 3.57 | 3.64 | 35.75 |
| | SynthID | 8.9 | 0.57 | 3.50 | 3.77 | 3.01 | 4.10 | 0.96 | 3.07 | 36.26 | 36.26 | ∞ | 20.48 | 3.94 | 3.61 | 3.66 | 36.14 |
| | EXP | 33.8 | 0.38 | 3.11 | 3.15 | 2.43 | 3.89 | 0.88 | 2.56 | 23.93 | 23.93 | ∞ | 10.77 | 3.14 | 3.09 | 3.21 | 23.73 |
| | WavMark | 9.8 | 0.58 | 3.50 | 3.81 | 2.99 | 4.08 | 0.96 | 2.97 | 35.86 | 35.86 | ∞ | 20.11 | 3.81 | 3.59 | 3.70 | 35.81 |
| | Timbre | 9.8 | 0.57 | 3.42 | 3.59 | 2.86 | 4.08 | 0.95 | 2.77 | 34.02 | 34.02 | 33.80 | 18.85 | 3.78 | 3.50 | 3.67 | 33.95 |
| | AudioSeal | 10.0 | 0.57 | 3.56 | 3.79 | 2.99 | 4.08 | 0.96 | 3.01 | 36.08 | 36.08 | ∞ | 19.54 | 3.98 | 3.59 | 3.63 | 35.88 |
| | SilentCipher | 9.8 | 0.53 | 3.56 | 4.06 | 3.01 | 4.09 | 0.95 | 3.10 | 36.34 | 36.34 | ∞ | 20.49 | 4.00 | 3.59 | 3.67 | 35.81 |

In-processing.  Post-processing.  ↑ Higher is better.  ↓ Lower is better.  ∞ Infinity.

Table 12: Quality comparison across watermarking methods on CV3-Eval Chinese.

| Model | Method | WER↓ | SIM↑ | UT↑ | PLC↑ | DNS↑ | MOS↑ | STOI↑ | PESQ↑ | SAR↑ | SDR↑ | SNR↑ | Si-SDR↑ | SSQA↑ | DNS-P↑ | SING↑ | Ci-SDR↑ |
|---|---|---|---|---|---|---|---|---|---|---|---|---|---|---|---|---|---|
| FireRedTTS | No Watermark | 4.4 | 0.68 | 2.55 | 4.10 | 3.12 | 4.19 | 0.98 | 3.04 | 32.86 | 32.86 | ∞ | 20.91 | 4.05 | 3.75 | 3.82 | 32.56 |
| | KGW | 4.5 | 0.68 | 2.55 | 4.09 | 3.12 | 4.20 | 0.97 | 3.04 | 32.37 | 32.37 | 32.45 | 20.88 | 4.03 | 3.74 | 3.81 | 32.24 |
| | Unigram | 4.6 | 0.67 | 2.54 | 4.10 | 3.12 | 4.19 | 0.98 | 3.04 | 32.51 | 32.51 | ∞ | 20.79 | 4.02 | 3.74 | 3.81 | 32.33 |
| | SWEET | 4.7 | 0.68 | 2.55 | 4.09 | 3.12 | 4.21 | 0.97 | 3.03 | 32.94 | 32.94 | 33.07 | 20.90 | 4.04 | 3.75 | 3.80 | 32.77 |
| | MorphMark | 4.1 | 0.68 | 2.56 | 4.11 | 3.12 | 4.18 | 0.98 | 3.05 | 32.84 | 32.84 | 32.95 | 20.98 | 4.07 | 3.74 | 3.81 | 32.75 |
| | SynthID | 4.7 | 0.68 | 2.53 | 4.09 | 3.11 | 4.20 | 0.98 | 3.04 | 32.72 | 32.72 | 32.78 | 20.92 | 4.04 | 3.75 | 3.80 | 32.53 |
| | EXP | 3.9 | 0.70 | 3.08 | 4.24 | 3.20 | 4.15 | 0.98 | 3.27 | 34.28 | 34.28 | ∞ | 22.80 | 4.21 | 3.77 | 3.85 | 34.15 |
| | WavMark | 4.5 | 0.69 | 2.50 | 4.05 | 3.10 | 4.20 | 0.97 | 2.90 | 32.43 | 32.43 | ∞ | 20.52 | 3.78 | 3.72 | 3.83 | 32.31 |
| | Timbre | 4.4 | 0.67 | 2.44 | 3.92 | 3.02 | 4.15 | 0.97 | 2.79 | 31.76 | 31.76 | ∞ | 19.98 | 3.81 | 3.60 | 3.81 | 31.62 |
| | AudioSeal | 4.4 | 0.68 | 2.53 | 4.11 | 3.10 | 4.20 | 0.97 | 2.99 | 32.59 | 32.59 | ∞ | 20.44 | 4.00 | 3.72 | 3.78 | 32.41 |
| | SilentCipher | 4.5 | 0.68 | 2.55 | 4.12 | 3.12 | 4.20 | 0.98 | 3.06 | 33.13 | 33.13 | ∞ | 20.96 | 4.08 | 3.76 | 3.80 | 32.81 |
| Fish-Speech | No Watermark | 4.0 | 0.59 | 3.22 | 4.43 | 3.22 | 4.02 | 0.98 | 3.41 | 38.90 | 38.90 | 38.54 | 23.56 | 4.42 | 3.84 | 4.06 | 38.67 |
| | KGW | 5.0 | 0.59 | 3.19 | 4.42 | 3.21 | 4.03 | 0.98 | 3.39 | 39.24 | 39.24 | 38.89 | 23.41 | 4.41 | 3.84 | 4.05 | 38.96 |
| | Unigram | 4.2 | 0.59 | 3.21 | 4.42 | 3.22 | 4.02 | 0.99 | 3.39 | 38.82 | 38.82 | 38.46 | 23.49 | 4.41 | 3.85 | 4.06 | 38.58 |
| | SWEET | 4.1 | 0.59 | 3.19 | 4.41 | 3.21 | 4.02 | 0.98 | 3.39 | 38.81 | 38.81 | 38.48 | 23.51 | 4.39 | 3.84 | 4.07 | 38.56 |
| | MorphMark | 4.1 | 0.59 | 3.22 | 4.43 | 3.21 | 4.03 | 0.99 | 3.41 | 39.09 | 39.09 | 38.74 | 23.55 | 4.43 | 3.84 | 4.06 | 38.91 |
| | SynthID | 4.1 | 0.59 | 3.18 | 4.42 | 3.20 | 4.02 | 0.98 | 3.40 | 38.89 | 38.89 | 38.55 | 23.50 | 4.40 | 3.83 | 4.06 | 38.69 |
| | EXP | 3.7 | 0.61 | 3.42 | 4.45 | 3.25 | 4.03 | 0.99 | 3.51 | 39.24 | 39.24 | 38.85 | 24.29 | 4.48 | 3.83 | 4.10 | 38.84 |
| | WavMark | 4.0 | 0.60 | 3.11 | 4.32 | 3.13 | 3.97 | 0.98 | 3.14 | 37.81 | 37.81 | 37.56 | 22.93 | 4.16 | 3.79 | 3.99 | 37.60 |
| | Timbre | 4.0 | 0.57 | 3.08 | 4.21 | 3.15 | 4.08 | 0.98 | 3.22 | 38.21 | 38.21 | 37.95 | 22.72 | 4.29 | 3.73 | 3.99 | 37.98 |
| | AudioSeal | 3.9 | 0.59 | 3.19 | 4.42 | 3.18 | 4.03 | 0.98 | 3.35 | 38.48 | 38.48 | 38.13 | 22.93 | 4.39 | 3.82 | 4.05 | 38.17 |
| | SilentCipher | 4.0 | 0.59 | 3.22 | 4.45 | 3.23 | 4.05 | 0.98 | 3.43 | 39.04 | 39.04 | 38.67 | 23.67 | 4.43 | 3.85 | 4.06 | 38.78 |
| Spark-TTS | No Watermark | 5.9 | 0.68 | 2.96 | 4.00 | 3.15 | 4.04 | 0.98 | 3.17 | 36.49 | 36.49 | ∞ | 21.94 | 4.24 | 3.69 | 3.82 | 36.37 |
| | KGW | 5.1 | 0.68 | 2.92 | 4.01 | 3.15 | 4.07 | 0.97 | 3.17 | 36.88 | 36.88 | ∞ | 22.00 | 4.24 | 3.70 | 3.81 | 36.68 |
| | Unigram | 9.0 | 0.66 | 2.91 | 3.92 | 3.10 | 4.06 | 0.97 | 3.09 | 35.83 | 35.83 | ∞ | 20.81 | 4.15 | 3.64 | 3.74 | 35.61 |
| | SWEET | 5.6 | 0.68 | 2.93 | 3.99 | 3.15 | 4.07 | 0.97 | 3.14 | 36.47 | 36.47 | ∞ | 21.71 | 4.24 | 3.69 | 3.80 | 36.34 |
| | MorphMark | 5.1 | 0.68 | 2.96 | 4.01 | 3.15 | 4.05 | 0.98 | 3.16 | 36.54 | 36.54 | ∞ | 21.88 | 4.25 | 3.69 | 3.80 | 36.32 |
| | SynthID | 4.8 | 0.69 | 2.92 | 4.00 | 3.15 | 4.07 | 0.98 | 3.15 | 36.99 | 36.99 | 37.16 | 21.93 | 4.26 | 3.70 | 3.82 | 36.81 |
| | EXP | 31.7 | 0.46 | 2.65 | 3.41 | 2.61 | 3.90 | 0.91 | 2.74 | 27.45 | 27.45 | ∞ | 14.75 | 3.45 | 3.19 | 3.28 | 27.32 |
| | WavMark | 5.7 | 0.69 | 2.87 | 3.96 | 3.13 | 4.09 | 0.97 | 3.05 | 36.28 | 36.28 | ∞ | 21.68 | 4.03 | 3.67 | 3.83 | 36.26 |
| | Timbre | 6.1 | 0.68 | 2.81 | 3.69 | 3.00 | 4.05 | 0.97 | 2.92 | 33.94 | 33.94 | 33.91 | 21.01 | 4.09 | 3.59 | 3.78 | 33.79 |
| | AudioSeal | 8.4 | 0.68 | 2.94 | 3.96 | 3.13 | 4.03 | 0.98 | 3.11 | 36.34 | 36.34 | ∞ | 21.24 | 4.22 | 3.68 | 3.78 | 36.00 |
| | SilentCipher | 7.7 | 0.62 | 2.97 | 4.16 | 3.13 | 4.05 | 0.97 | 3.17 | 36.60 | 36.60 | ∞ | 21.67 | 4.25 | 3.68 | 3.80 | 36.32 |

■ In-processing. ■ Post-processing. ↑ Higher is better. ↓ Lower is better. ∞ Infinity.

Table 13: Robustness evaluation results on Seed-TTS-Eval under attacks. All metrics represent TPR@3.0% FPR (higher is better).

| Model | Method | EN TS | SMH | GN | BN | Echo | MP3 | ECD | QNT | HPF | LPF | ZH TS | SMH | GN | BN | Echo | MP3 | ECD | QNT | HPF | LPF |
|---|---|---|---|---|---|---|---|---|---|---|---|---|---|---|---|---|---|---|---|---|---|
| FireRedTTS | KGW | 0.000 | 0.054 | 0.392 | 0.528 | 0.002 | 0.954 | 0.122 | 0.070 | 0.000 | 0.966 | 0.000 | 0.071 | 0.823 | 0.939 | 0.002 | 1.000 | 0.183 | 0.139 | 0.000 | 1.000 |
| | Unigram | 0.307 | 0.232 | 0.891 | 0.964 | 0.090 | 1.000 | 0.658 | 0.244 | 1.000 | 1.000 | 0.503 | 0.389 | 0.999 | 1.000 | 0.358 | 1.000 | 0.893 | 0.570 | 1.000 | 1.000 |
| | SWEET | 0.000 | 0.057 | 0.381 | 0.562 | 0.017 | 0.962 | 0.121 | 0.061 | 0.000 | 0.976 | 0.000 | 0.056 | 0.822 | 0.937 | 0.035 | 1.000 | 0.190 | 0.118 | 0.000 | 1.000 |
| | MorphMark | 0.000 | 0.025 | 0.100 | 0.100 | 0.000 | 0.244 | 0.050 | 0.063 | 0.000 | 0.274 | 0.000 | 0.037 | 0.132 | 0.178 | 0.000 | 0.339 | 0.051 | 0.062 | 0.000 | 0.360 |
| | SynthID | 0.082 | 0.069 | 0.505 | 0.676 | 0.235 | 0.994 | 0.140 | 0.085 | 0.000 | 0.995 | 0.072 | 0.054 | 0.785 | 0.923 | 0.237 | 1.000 | 0.176 | 0.114 | 0.000 | 1.000 |
| | EXP | 0.153 | 0.346 | 0.966 | 0.993 | 0.615 | 1.000 | 0.803 | 0.244 | 1.000 | 1.000 | 0.102 | 0.346 | 1.000 | 1.000 | 0.767 | 1.000 | 0.829 | 0.564 | 0.000 | 1.000 |
| | WavMark | 0.506 | 1.000 | 0.987 | 1.000 | 0.976 | 1.000 | 0.000 | 0.159 | 0.000 | 1.000 | 0.770 | 1.000 | 1.000 | 1.000 | 1.000 | 1.000 | 0.000 | 0.408 | 0.000 | 1.000 |
| | Timbre | 1.000 | 1.000 | 1.000 | 1.000 | 0.999 | 1.000 | 0.523 | 0.998 | 0.000 | 1.000 | 1.000 | 1.000 | 1.000 | 1.000 | 1.000 | 1.000 | 0.451 | 1.000 | 0.000 | 1.000 |
| | AudioSeal | 0.993 | 0.999 | 1.000 | 1.000 | 0.999 | 1.000 | 0.829 | 0.988 | 0.000 | 1.000 | 0.973 | 1.000 | 1.000 | 1.000 | 1.000 | 1.000 | 0.806 | 0.993 | 0.000 | 1.000 |
| | SilentCipher | 0.000 | 0.603 | 0.867 | 0.807 | 0.479 | 0.981 | 0.000 | 0.000 | 0.000 | 0.980 | 0.000 | 0.824 | 0.988 | 0.984 | 0.797 | 0.998 | 0.000 | 0.000 | 0.000 | 0.998 |
| Fish-Speech | KGW | 0.000 | 0.000 | 0.000 | 0.002 | 0.000 | 0.251 | 0.000 | 0.000 | 0.000 | 0.234 | 0.000 | 0.000 | 0.004 | 0.018 | 0.000 | 0.560 | 0.000 | 0.000 | 0.000 | 0.550 |
| | Unigram | 0.541 | 0.244 | 0.099 | 0.327 | 0.107 | 0.989 | 0.096 | 0.294 | 0.013 | 0.985 | 0.716 | 0.329 | 0.340 | 0.696 | 0.074 | 0.998 | 0.159 | 0.329 | 0.007 | 0.998 |
| | SWEET | 0.000 | 0.001 | 0.000 | 0.001 | 0.000 | 0.218 | 0.000 | 0.000 | 0.000 | 0.240 | 0.000 | 0.000 | 0.002 | 0.010 | 0.000 | 0.539 | 0.000 | 0.000 | 0.000 | 0.547 |
| | MorphMark | 0.000 | 0.000 | 0.001 | 0.000 | 0.000 | 0.002 | 0.000 | 0.000 | 0.000 | 0.001 | 0.000 | 0.000 | 0.000 | 0.000 | 0.000 | 0.003 | 0.000 | 0.000 | 0.000 | 0.001 |
| | SynthID | 0.014 | 0.091 | 0.089 | 0.150 | 0.026 | 0.824 | 0.070 | 0.031 | 0.000 | 0.818 | 0.005 | 0.095 | 0.159 | 0.315 | 0.034 | 0.951 | 0.092 | 0.028 | 0.000 | 0.957 |
| | EXP | 0.000 | 0.255 | 0.148 | 0.335 | 0.013 | 0.979 | 0.068 | 0.003 | 0.000 | 0.979 | 0.000 | 0.219 | 0.425 | 0.652 | 0.026 | 0.996 | 0.104 | 0.008 | 0.000 | 0.996 |
| | WavMark | 0.936 | 0.994 | 0.765 | 0.955 | 0.982 | 1.000 | 0.000 | 0.105 | 0.000 | 1.000 | 0.980 | 1.000 | 0.956 | 0.999 | 1.000 | 1.000 | 0.000 | 0.205 | 0.000 | 1.000 |
| | Timbre | 1.000 | 1.000 | 1.000 | 1.000 | 0.994 | 1.000 | 0.483 | 0.999 | 0.000 | 1.000 | 1.000 | 1.000 | 1.000 | 1.000 | 1.000 | 1.000 | 0.596 | 1.000 | 0.000 | 1.000 |
| | AudioSeal | 0.993 | 1.000 | 1.000 | 1.000 | 0.997 | 1.000 | 1.000 | 0.997 | 0.000 | 1.000 | 0.973 | 1.000 | 1.000 | 1.000 | 0.999 | 1.000 | 1.000 | 0.998 | 0.000 | 1.000 |
| | SilentCipher | 0.000 | 0.767 | 0.907 | 0.845 | 0.417 | 0.965 | 0.000 | 0.002 | 0.000 | 0.967 | 0.000 | 0.923 | 0.986 | 0.983 | 0.694 | 0.999 | 0.000 | 0.005 | 0.000 | 0.999 |
| Spark-TTS | KGW | 0.175 | 0.042 | 0.454 | 0.579 | 0.197 | 0.666 | 0.155 | 0.056 | 0.000 | 0.686 | 0.215 | 0.043 | 0.552 | 0.659 | 0.155 | 0.761 | 0.155 | 0.064 | 0.000 | 0.766 |
| | Unigram | 0.501 | 0.137 | 0.849 | 0.915 | 0.201 | 0.926 | 0.647 | 0.170 | 1.000 | 0.926 | 0.540 | 0.134 | 0.964 | 0.986 | 0.099 | 0.988 | 0.841 | 0.272 | 1.000 | 0.991 |
| | SWEET | 0.412 | 0.056 | 0.481 | 0.571 | 0.142 | 0.683 | 0.156 | 0.069 | 0.014 | 0.679 | 0.635 | 0.059 | 0.597 | 0.695 | 0.080 | 0.762 | 0.188 | 0.069 | 0.005 | 0.765 |
| | MorphMark | 0.174 | 0.048 | 0.113 | 0.149 | 0.184 | 0.179 | 0.086 | 0.040 | 0.000 | 0.181 | 0.244 | 0.034 | 0.149 | 0.185 | 0.104 | 0.223 | 0.074 | 0.027 | 0.000 | 0.224 |
| | SynthID | 0.079 | 0.032 | 0.413 | 0.531 | 0.029 | 0.628 | 0.153 | 0.040 | 1.000 | 0.627 | 0.077 | 0.042 | 0.573 | 0.667 | 0.056 | 0.772 | 0.177 | 0.075 | 1.000 | 0.778 |
| | EXP | 0.045 | 0.099 | 0.673 | 0.733 | 0.188 | 0.739 | 0.473 | 0.144 | 0.003 | 0.744 | 0.060 | 0.099 | 0.893 | 0.908 | 0.231 | 0.912 | 0.674 | 0.257 | 0.000 | 0.919 |
| | WavMark | 0.179 | 0.997 | 0.993 | 0.997 | 0.948 | 1.000 | 0.000 | 0.057 | 0.000 | 0.997 | 0.321 | 0.996 | 0.995 | 0.996 | 0.992 | 1.000 | 0.000 | 0.115 | 0.000 | 0.996 |
| | Timbre | 0.999 | 1.000 | 1.000 | 1.000 | 1.000 | 1.000 | 0.402 | 0.997 | 0.000 | 1.000 | 1.000 | 1.000 | 1.000 | 1.000 | 1.000 | 1.000 | 0.225 | 0.997 | 0.000 | 1.000 |
| | AudioSeal | 0.980 | 0.994 | 1.000 | 1.000 | 0.999 | 1.000 | 0.000 | 0.902 | 0.000 | 1.000 | 0.965 | 0.996 | 1.000 | 1.000 | 1.000 | 1.000 | 0.085 | 0.950 | 0.000 | 1.000 |
| | SilentCipher | 0.000 | 0.488 | 0.518 | 0.824 | 0.360 | 0.953 | 0.000 | 0.000 | 0.001 | 0.958 | 0.000 | 0.708 | 0.850 | 0.974 | 0.617 | 0.990 | 0.000 | 0.000 | 0.000 | 0.991 |

■ In-processing. ■ Post-processing. Red Low robustness (< 0.3). Orange Medium robustness (0.3 − 0.5). Green High robustness (> 0.7).

Table 14: Robustness evaluation results on CV3-Eval under attacks. All metrics represent TPR@0.2% FPR (higher is better).

| Model | Method | EN | | | | | | | | | | ZH | | | | | | | | | |
|---|---|---|---|---|---|---|---|---|---|---|---|---|---|---|---|---|---|---|---|---|---|
| | | TS | SMH | GN | BN | Echo | MP3 | ECD | QNT | HPF | LPF | TS | SMH | GN | BN | Echo | MP3 | ECD | QNT | HPF | LPF |
| FireRedTTS | KGW | 0.000 | 0.006 | 0.438 | 0.270 | 0.006 | 0.908 | 0.036 | 0.174 | 0.000 | 0.980 | 0.000 | 0.010 | 0.452 | 0.788 | 0.022 | 0.988 | 0.030 | 0.118 | 0.000 | 0.994 |
| | Unigram | 0.314 | 0.238 | 0.858 | 0.938 | 0.566 | 0.986 | 0.454 | 0.410 | 1.000 | 1.000 | 0.358 | 0.378 | 0.908 | 0.976 | 0.648 | 1.000 | 0.726 | 0.398 | 1.000 | 1.000 |
| | SWEET | 0.000 | 0.008 | 0.416 | 0.255 | 0.034 | 0.908 | 0.086 | 0.096 | 0.000 | 0.988 | 0.000 | 0.004 | 0.415 | 0.772 | 0.060 | 0.988 | 0.064 | 0.052 | 0.000 | 0.998 |
| | MorphMark | 0.000 | 0.000 | 0.016 | 0.026 | 0.000 | 0.100 | 0.004 | 0.146 | 0.000 | 0.128 | 0.000 | 0.002 | 0.064 | 0.112 | 0.000 | 0.395 | 0.008 | 0.104 | 0.000 | 0.437 |
| | SynthID | 0.026 | 0.014 | 0.378 | 0.342 | 0.182 | 0.958 | 0.028 | 0.188 | 0.000 | 0.984 | 0.028 | 0.016 | 0.479 | 0.663 | 0.251 | 0.992 | 0.052 | 0.140 | 0.000 | 0.996 |
| | EXP | 0.106 | 0.096 | 0.838 | 0.934 | 0.748 | 0.996 | 0.384 | 0.134 | 0.000 | 1.000 | 0.088 | 0.118 | 0.854 | 0.972 | 0.794 | 1.000 | 0.569 | 0.146 | 0.000 | 1.000 |
| | WavMark | 0.800 | 0.974 | 0.976 | 0.990 | 0.988 | 0.990 | 0.000 | 0.235 | 0.000 | 0.990 | 0.862 | 1.000 | 0.988 | 1.000 | 1.000 | 1.000 | 0.000 | 0.289 | 0.000 | 1.000 |
| | Timbre | 1.000 | 1.000 | 1.000 | 1.000 | 1.000 | 1.000 | 0.424 | 0.998 | 0.000 | 1.000 | 1.000 | 1.000 | 1.000 | 0.998 | 1.000 | 1.000 | 0.533 | 1.000 | 0.000 | 1.000 |
| | AudioSeal | 0.890 | 0.942 | 1.000 | 1.000 | 0.996 | 1.000 | 0.752 | 0.954 | 0.000 | 1.000 | 0.882 | 0.996 | 1.000 | 0.998 | 1.000 | 1.000 | 0.852 | 0.988 | 0.000 | 1.000 |
| | SilentCipher | 0.000 | 0.868 | 0.898 | 0.874 | 0.872 | 0.982 | 0.010 | 0.000 | 0.098 | 0.984 | 0.000 | 0.932 | 0.970 | 0.966 | 0.966 | 1.000 | 0.008 | 0.004 | 0.024 | 1.000 |
| Fish-Speech | KGW | 0.000 | 0.000 | 0.000 | 0.002 | 0.000 | 0.130 | 0.000 | 0.000 | 0.000 | 0.128 | 0.000 | 0.000 | 0.000 | 0.002 | 0.000 | 0.261 | 0.000 | 0.000 | 0.000 | 0.257 |
| | Unigram | 0.346 | 0.180 | 0.130 | 0.236 | 0.028 | 0.960 | 0.045 | 0.108 | 0.110 | 0.960 | 0.494 | 0.350 | 0.138 | 0.220 | 0.032 | 0.990 | 0.063 | 0.136 | 0.132 | 0.990 |
| | SWEET | 0.000 | 0.000 | 0.000 | 0.000 | 0.000 | 0.146 | 0.000 | 0.000 | 0.000 | 0.132 | 0.000 | 0.000 | 0.000 | 0.000 | 0.000 | 0.254 | 0.000 | 0.000 | 0.000 | 0.266 |
| | MorphMark | 0.000 | 0.000 | 0.000 | 0.000 | 0.000 | 0.000 | 0.000 | 0.000 | 0.000 | 0.000 | 0.000 | 0.000 | 0.000 | 0.000 | 0.000 | 0.000 | 0.000 | 0.000 | 0.000 | 0.000 |
| | SynthID | 0.004 | 0.028 | 0.004 | 0.034 | 0.006 | 0.570 | 0.004 | 0.006 | 0.000 | 0.584 | 0.000 | 0.032 | 0.042 | 0.062 | 0.000 | 0.778 | 0.002 | 0.008 | 0.000 | 0.796 |
| | EXP | 0.000 | 0.056 | 0.094 | 0.138 | 0.000 | 0.892 | 0.000 | 0.000 | 0.000 | 0.888 | 0.000 | 0.124 | 0.110 | 0.124 | 0.000 | 0.972 | 0.014 | 0.000 | 0.000 | 0.978 |
| | WavMark | 0.974 | 0.971 | 0.640 | 0.926 | 0.998 | 1.000 | 0.000 | 0.044 | 0.000 | 1.000 | 0.996 | 0.996 | 0.546 | 0.942 | 1.000 | 1.000 | 0.000 | 0.052 | 0.000 | 1.000 |
| | Timbre | 1.000 | 1.000 | 1.000 | 1.000 | 1.000 | 1.000 | 0.493 | 1.000 | 0.000 | 1.000 | 1.000 | 1.000 | 1.000 | 1.000 | 1.000 | 1.000 | 0.604 | 0.998 | 0.000 | 1.000 |
| | AudioSeal | 0.944 | 1.000 | 1.000 | 1.000 | 0.994 | 1.000 | 0.988 | 0.992 | 0.000 | 1.000 | 0.910 | 1.000 | 1.000 | 1.000 | 0.998 | 1.000 | 0.998 | 0.996 | 0.000 | 1.000 |
| | SilentCipher | 0.000 | 0.812 | 0.868 | 0.826 | 0.600 | 0.968 | 0.000 | 0.000 | 0.000 | 0.964 | 0.000 | 0.956 | 0.976 | 0.962 | 0.778 | 0.998 | 0.000 | 0.004 | 0.000 | 0.998 |
| Spark-TTS | KGW | 0.034 | 0.004 | 0.292 | 0.338 | 0.028 | 0.408 | 0.077 | 0.024 | 0.000 | 0.406 | 0.052 | 0.004 | 0.320 | 0.452 | 0.054 | 0.524 | 0.074 | 0.010 | 0.000 | 0.524 |
| | Unigram | 0.314 | 0.116 | 0.630 | 0.728 | 0.164 | 0.806 | 0.514 | 0.210 | 0.998 | 0.816 | 0.320 | 0.082 | 0.844 | 0.896 | 0.104 | 0.954 | 0.586 | 0.166 | 0.998 | 0.952 |
| | SWEET | 0.302 | 0.016 | 0.294 | 0.356 | 0.050 | 0.408 | 0.042 | 0.046 | 0.048 | 0.412 | 0.436 | 0.014 | 0.358 | 0.440 | 0.040 | 0.572 | 0.067 | 0.036 | 0.020 | 0.568 |
| | MorphMark | 0.036 | 0.016 | 0.026 | 0.036 | 0.024 | 0.060 | 0.022 | 0.020 | 0.000 | 0.062 | 0.050 | 0.008 | 0.032 | 0.038 | 0.014 | 0.058 | 0.020 | 0.008 | 0.000 | 0.062 |
| | SynthID | 0.018 | 0.018 | 0.000 | 0.000 | 0.014 | 0.014 | 0.000 | 0.018 | 0.848 | 0.012 | 0.006 | 0.006 | 0.000 | 0.000 | 0.006 | 0.006 | 0.000 | 0.006 | 1.000 | 0.006 |
| | EXP | 0.052 | 0.118 | 0.446 | 0.490 | 0.182 | 0.536 | 0.224 | 0.146 | 0.000 | 0.518 | 0.050 | 0.102 | 0.534 | 0.618 | 0.202 | 0.638 | 0.238 | 0.124 | 0.000 | 0.650 |
| | WavMark | 0.471 | 0.986 | 0.974 | 0.986 | 0.962 | 0.986 | 0.000 | 0.104 | 0.000 | 0.986 | 0.529 | 0.996 | 0.992 | 0.996 | 0.994 | 0.996 | 0.000 | 0.128 | 0.000 | 0.996 |
| | Timbre | 1.000 | 1.000 | 1.000 | 0.996 | 1.000 | 1.000 | 0.327 | 0.980 | 0.000 | 1.000 | 1.000 | 1.000 | 1.000 | 0.996 | 1.000 | 1.000 | 0.481 | 0.994 | 0.000 | 1.000 |
| | AudioSeal | 0.866 | 0.968 | 1.000 | 1.000 | 1.000 | 1.000 | 0.018 | 0.876 | 0.000 | 1.000 | 0.820 | 0.990 | 1.000 | 1.000 | 1.000 | 1.000 | 0.020 | 0.852 | 0.000 | 1.000 |
| | SilentCipher | 0.000 | 0.708 | 0.648 | 0.848 | 0.639 | 0.914 | 0.000 | 0.000 | 0.062 | 0.928 | 0.000 | 0.790 | 0.806 | 0.960 | 0.772 | 0.978 | 0.000 | 0.000 | 0.012 | 0.982 |

In-processing. Post-processing. Red Low robustness (< 0.3). Orange Medium robustness (0.3 − 0.5). Green High robustness (> 0.7).

Table 15: Robustness evaluation results on CV3-Eval under attacks. All metrics represent TPR@3.0% FPR (higher is better).

| Model | Method | EN | | | | | | | | | | ZH | | | | | | | | | |
|---|---|---|---|---|---|---|---|---|---|---|---|---|---|---|---|---|---|---|---|---|---|
| | | TS | SMH | GN | BN | Echo | MP3 | ECD | QNT | HPF | LPF | TS | SMH | GN | BN | Echo | MP3 | ECD | QNT | HPF | LPF |
| FireRedTTS | KGW | 0.000 | 0.068 | 0.620 | 0.716 | 0.068 | 0.958 | 0.152 | 0.268 | 0.000 | 0.990 | 0.000 | 0.066 | 0.658 | 0.802 | 0.064 | 0.998 | 0.182 | 0.234 | 0.000 | 0.998 |
| | Unigram | 0.550 | 0.508 | 0.916 | 0.966 | 0.706 | 0.986 | 0.622 | 0.594 | 1.000 | 1.000 | 0.630 | 0.634 | 0.960 | 0.990 | 0.828 | 1.000 | 0.858 | 0.582 | 1.000 | 1.000 |
| | SWEET | 0.000 | 0.080 | 0.612 | 0.694 | 0.142 | 0.964 | 0.240 | 0.172 | 0.000 | 0.996 | 0.000 | 0.072 | 0.659 | 0.838 | 0.214 | 0.996 | 0.218 | 0.148 | 0.000 | 1.000 |
| | MorphMark | 0.000 | 0.036 | 0.142 | 0.136 | 0.000 | 0.328 | 0.026 | 0.226 | 0.000 | 0.394 | 0.000 | 0.040 | 0.208 | 0.295 | 0.004 | 0.629 | 0.058 | 0.194 | 0.000 | 0.673 |
| | SynthID | 0.058 | 0.050 | 0.496 | 0.706 | 0.348 | 0.994 | 0.102 | 0.278 | 0.000 | 0.996 | 0.074 | 0.050 | 0.617 | 0.802 | 0.483 | 0.998 | 0.178 | 0.255 | 0.000 | 1.000 |
| | EXP | 0.194 | 0.328 | 0.916 | 0.974 | 0.872 | 0.998 | 0.590 | 0.280 | 0.000 | 1.000 | 0.142 | 0.345 | 0.936 | 0.986 | 0.924 | 1.000 | 0.800 | 0.299 | 0.000 | 1.000 |
| | WavMark | 0.800 | 0.974 | 0.976 | 0.990 | 0.988 | 0.990 | 0.000 | 0.235 | 0.000 | 0.990 | 0.862 | 1.000 | 0.988 | 1.000 | 1.000 | 1.000 | 0.000 | 0.289 | 0.000 | 1.000 |
| | Timbre | 1.000 | 1.000 | 1.000 | 1.000 | 1.000 | 1.000 | 0.424 | 0.998 | 0.000 | 1.000 | 1.000 | 1.000 | 1.000 | 0.998 | 1.000 | 1.000 | 0.533 | 1.000 | 0.000 | 1.000 |
| | AudioSeal | 0.890 | 0.942 | 1.000 | 1.000 | 0.996 | 1.000 | 0.752 | 0.954 | 0.000 | 1.000 | 0.882 | 0.996 | 1.000 | 0.998 | 1.000 | 1.000 | 0.852 | 0.988 | 0.000 | 1.000 |
| | SilentCipher | 0.000 | 0.868 | 0.904 | 0.874 | 0.876 | 0.984 | 0.010 | 0.002 | 0.102 | 0.984 | 0.000 | 0.932 | 0.970 | 0.966 | 0.966 | 1.000 | 0.010 | 0.004 | 0.026 | 1.000 |
| Fish-Speech | KGW | 0.000 | 0.002 | 0.000 | 0.002 | 0.000 | 0.334 | 0.000 | 0.000 | 0.000 | 0.344 | 0.000 | 0.000 | 0.000 | 0.002 | 0.000 | 0.505 | 0.000 | 0.000 | 0.000 | 0.519 |
| | Unigram | 0.678 | 0.448 | 0.226 | 0.368 | 0.128 | 0.984 | 0.153 | 0.316 | 0.156 | 0.976 | 0.796 | 0.644 | 0.232 | 0.376 | 0.140 | 0.996 | 0.176 | 0.400 | 0.212 | 0.996 |
| | SWEET | 0.000 | 0.000 | 0.002 | 0.008 | 0.000 | 0.298 | 0.000 | 0.000 | 0.000 | 0.330 | 0.000 | 0.000 | 0.002 | 0.006 | 0.000 | 0.518 | 0.000 | 0.000 | 0.000 | 0.492 |
| | MorphMark | 0.000 | 0.000 | 0.000 | 0.000 | 0.000 | 0.002 | 0.000 | 0.000 | 0.000 | 0.000 | 0.000 | 0.000 | 0.000 | 0.000 | 0.000 | 0.000 | 0.000 | 0.000 | 0.000 | 0.000 |
| | SynthID | 0.028 | 0.176 | 0.146 | 0.186 | 0.048 | 0.858 | 0.081 | 0.048 | 0.000 | 0.848 | 0.008 | 0.172 | 0.136 | 0.210 | 0.036 | 0.922 | 0.071 | 0.026 | 0.000 | 0.926 |
| | EXP | 0.002 | 0.184 | 0.184 | 0.240 | 0.008 | 0.944 | 0.032 | 0.002 | 0.000 | 0.944 | 0.002 | 0.308 | 0.172 | 0.206 | 0.012 | 0.984 | 0.066 | 0.000 | 0.000 | 0.986 |
| | WavMark | 0.974 | 0.971 | 0.640 | 0.926 | 0.998 | 1.000 | 0.000 | 0.044 | 0.000 | 1.000 | 0.996 | 0.996 | 0.546 | 0.942 | 1.000 | 1.000 | 0.000 | 0.052 | 0.000 | 1.000 |
| | Timbre | 1.000 | 1.000 | 1.000 | 1.000 | 1.000 | 1.000 | 0.493 | 1.000 | 0.000 | 1.000 | 1.000 | 1.000 | 1.000 | 1.000 | 1.000 | 1.000 | 0.604 | 0.998 | 0.000 | 1.000 |
| | AudioSeal | 0.944 | 1.000 | 1.000 | 1.000 | 0.994 | 1.000 | 0.988 | 0.992 | 0.000 | 1.000 | 0.910 | 1.000 | 1.000 | 1.000 | 0.998 | 1.000 | 0.998 | 0.996 | 0.000 | 1.000 |
| | SilentCipher | 0.000 | 0.812 | 0.868 | 0.828 | 0.604 | 0.968 | 0.000 | 0.000 | 0.000 | 0.964 | 0.000 | 0.956 | 0.976 | 0.962 | 0.780 | 0.998 | 0.000 | 0.006 | 0.000 | 0.998 |
| Spark-TTS | KGW | 0.106 | 0.054 | 0.486 | 0.546 | 0.210 | 0.614 | 0.253 | 0.072 | 0.000 | 0.614 | 0.126 | 0.038 | 0.600 | 0.678 | 0.234 | 0.772 | 0.231 | 0.056 | 0.000 | 0.768 |
| | Unigram | 0.534 | 0.246 | 0.788 | 0.858 | 0.319 | 0.890 | 0.705 | 0.330 | 0.998 | 0.896 | 0.536 | 0.190 | 0.952 | 0.956 | 0.246 | 0.984 | 0.817 | 0.336 | 1.000 | 0.980 |
| | SWEET | 0.448 | 0.060 | 0.490 | 0.550 | 0.180 | 0.614 | 0.216 | 0.120 | 0.048 | 0.622 | 0.676 | 0.056 | 0.608 | 0.698 | 0.164 | 0.770 | 0.200 | 0.136 | 0.020 | 0.778 |
| | MorphMark | 0.096 | 0.054 | 0.112 | 0.156 | 0.118 | 0.208 | 0.093 | 0.054 | 0.000 | 0.220 | 0.154 | 0.050 | 0.196 | 0.186 | 0.112 | 0.212 | 0.083 | 0.036 | 0.000 | 0.216 |
| | SynthID | 0.308 | 0.048 | 0.326 | 0.366 | 0.058 | 0.510 | 0.111 | 0.094 | 1.000 | 0.506 | 0.362 | 0.032 | 0.412 | 0.568 | 0.064 | 0.642 | 0.093 | 0.048 | 1.000 | 0.646 |
| | EXP | 0.118 | 0.148 | 0.550 | 0.584 | 0.294 | 0.598 | 0.368 | 0.210 | 0.000 | 0.586 | 0.130 | 0.148 | 0.630 | 0.702 | 0.312 | 0.696 | 0.412 | 0.212 | 0.000 | 0.698 |
| | WavMark | 0.471 | 0.986 | 0.974 | 0.986 | 0.962 | 0.986 | 0.000 | 0.104 | 0.000 | 0.986 | 0.529 | 0.996 | 0.992 | 0.996 | 0.994 | 0.996 | 0.000 | 0.128 | 0.000 | 0.996 |
| | Timbre | 1.000 | 1.000 | 1.000 | 0.996 | 1.000 | 1.000 | 0.327 | 0.980 | 0.000 | 1.000 | 1.000 | 1.000 | 1.000 | 0.996 | 1.000 | 1.000 | 0.481 | 0.994 | 0.000 | 1.000 |
| | AudioSeal | 0.866 | 0.968 | 1.000 | 1.000 | 1.000 | 1.000 | 0.018 | 0.876 | 0.000 | 1.000 | 0.820 | 0.990 | 1.000 | 1.000 | 1.000 | 1.000 | 0.020 | 0.852 | 0.000 | 1.000 |
| | SilentCipher | 0.000 | 0.712 | 0.660 | 0.848 | 0.649 | 0.918 | 0.000 | 0.000 | 0.064 | 0.930 | 0.000 | 0.808 | 0.812 | 0.960 | 0.780 | 0.978 | 0.000 | 0.000 | 0.012 | 0.982 |

In-processing. Post-processing. Red Low robustness (< 0.3). Orange Medium robustness (0.3 − 0.5). Green High robustness (> 0.7).

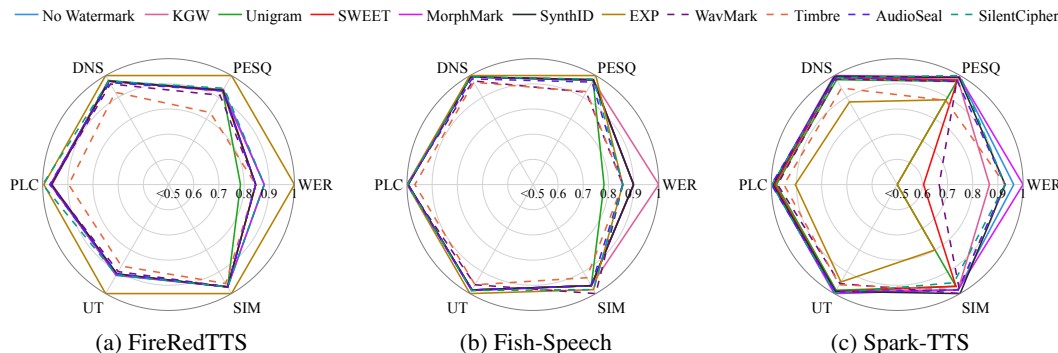

Figure 6: Quality results on Seed-TTS-Eval English. All metrics are normalized to the percentage of the best performance per metric. WER is inverted for consistent interpretation.

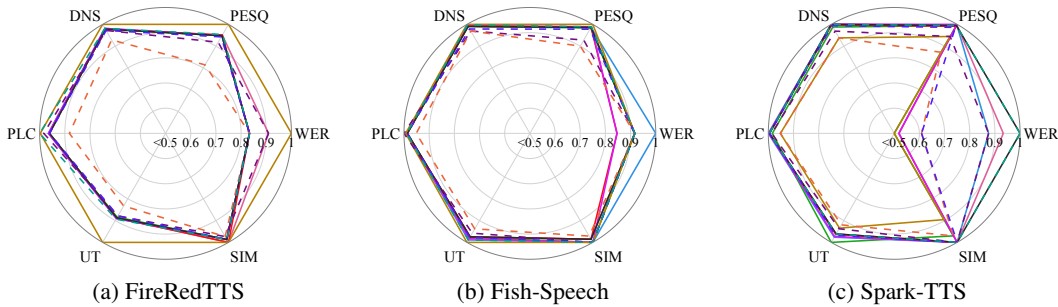

Figure 7: Quality results on Seed-TTS-Eval Chinese. All metrics are normalized to the percentage of the best performance per metric. WER is inverted for consistent interpretation.

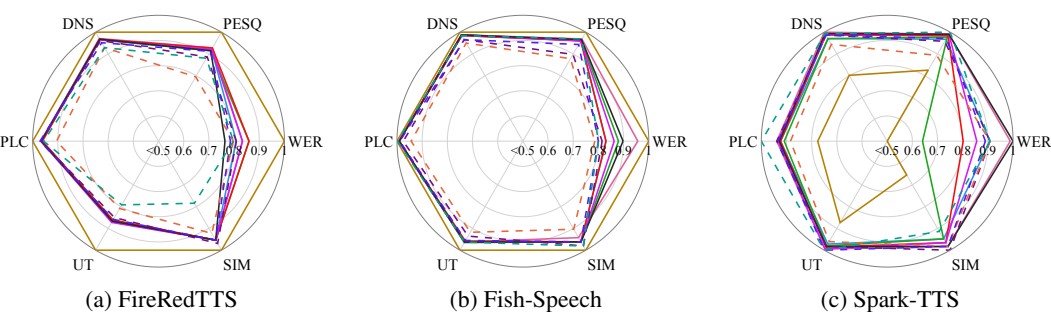

Figure 8: Quality results on CV3-Eval English. All metrics are normalized to the percentage of the best performance per metric. WER is inverted for consistent interpretation.

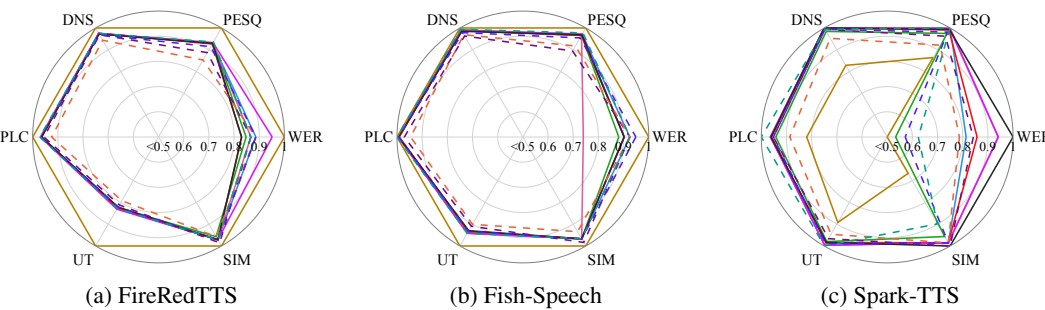

Figure 9: Quality results on CV3-Eval Chinese. All metrics are normalized to the percentage of the best performance per metric. WER is inverted for consistent interpretation.

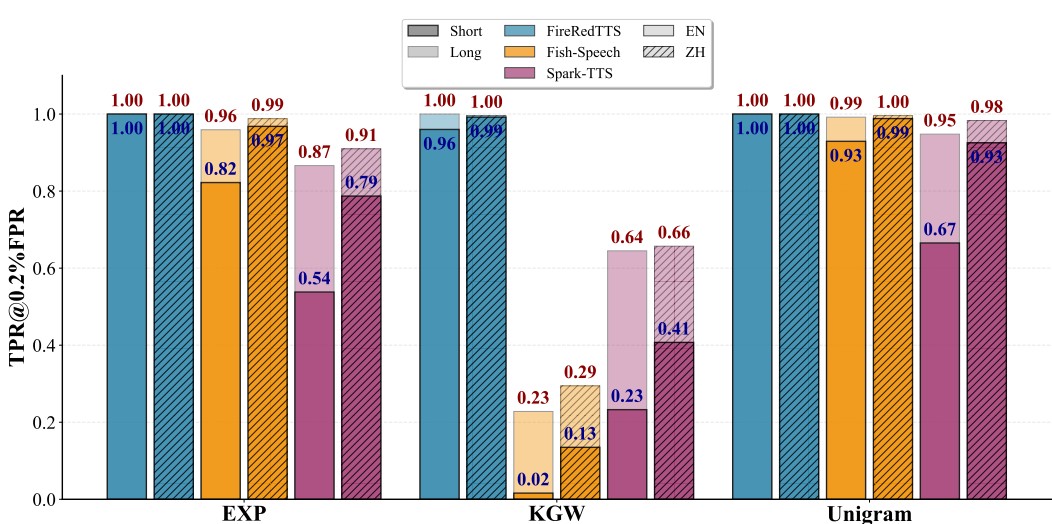

Figure 10: Token length effects on watermark detection of CV3-Eval dataset. Audio samples split by token count: short (0-50th percentile) vs. long (50th-100th percentile). Bars show TPR at fixed FPR, with darker bars for shorter sequences. Colors indicate TTS models (FireRedTTS: blue, Fish-Speech: orange, Spark-TTS: purple); hatching shows languages (solid: English, diagonal: Chinese). Longer sequences show better detectability across all methods.

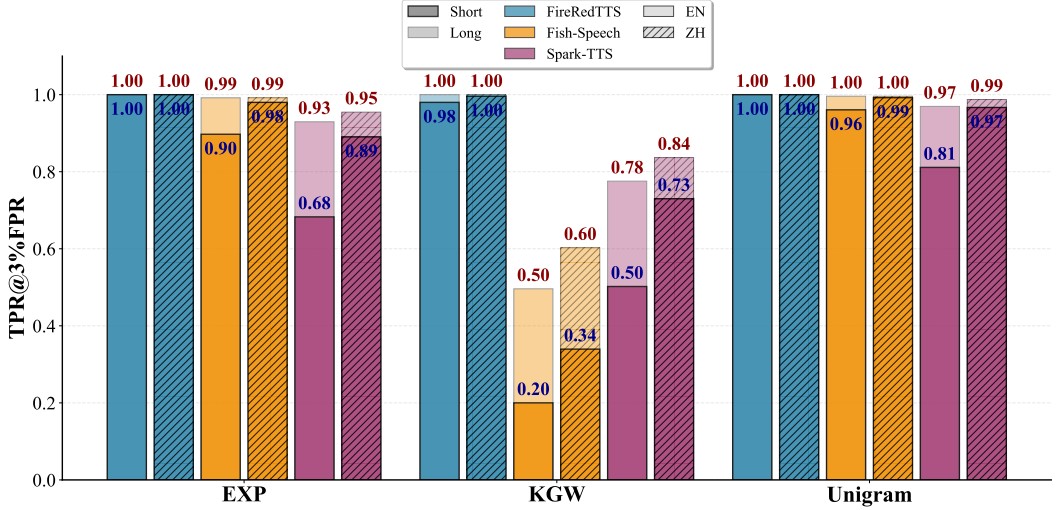

Figure 11: Token length effects on watermark detection of CV3-Eval dataset. Audio samples split by token count: short (0-50th percentile) vs. long (50th-100th percentile). Bars show TPR at fixed FPR, with darker bars for shorter sequences. Colors indicate TTS models (FireRedTTS: blue, Fish-Speech: orange, Spark-TTS: purple); hatching shows languages (solid: English, diagonal: Chinese). Longer sequences show better detectability across all methods.

