# OpenReview forum: "SpeechWakBench: How Well do Large Language Models Speak with Watermarks?"
_ICLR.cc/2026/Conference — ICLR 2026 Conference Withdrawn Submission_

### Official Review · Reviewer_jj8P · 2025-10-23

**Soundness:** 2
**Presentation:** 3
**Contribution:** 2
**Rating:** 4
**Confidence:** 5

**Summary:**

This paper presents a comprehensive evaluation framework for audio watermarking techniques, specifically focusing on Large Language Model (LLM)-based Text-to-Speech (TTS) systems. The authors conduct a systematic comparison between two primary watermarking paradigms: generation-time watermarking, adapted from the text domain, and post-processing watermarking. The study evaluates these methods across a suite of metrics, including audibility (quality), detectability, and robustness against a wide array of common audio attacks. The core finding of the paper is that while generation-time methods offer superior audio quality, they exhibit significant fragility and fail under most robustness tests. The authors attribute this failure to the difficulty of accurately reconstructing discrete speech tokens from perturbed audio waveforms, a necessary step for watermark detection in this paradigm.

**Strengths:**

1. This work is among the first to systematically apply and evaluate generation-time watermarking methods, originally developed for text, within the audio domain. The direct comparison against established post-processing audio watermarking techniques on key axes of quality, detectability, and robustness provides valuable insights for the field.

2. The paper is supported by extensive experiments covering multiple TTS models, watermarking algorithms, and a comprehensive set of 10 different audio attacks. The conclusion that the primary bottleneck for generation-time audio watermarking is the difficulty in accurately inverting audio waveforms back to discrete speech tokens is well-supported by the empirical results and provides a clear, actionable diagnosis of the problem.

**Weaknesses:**

1. Contribution as a "Benchmark": My primary concern lies with the paper's positioning as a benchmark. A benchmark typically involves the introduction of novel datasets, challenging evaluation protocols, or fundamentally new tasks. This work, while comprehensive, primarily integrates existing models, datasets, attack methodologies, and evaluation metrics into a unified framework. As such, its contribution feels more akin to a well-structured "toolkit" or a "report" of a large-scale empirical study rather than a foundational benchmark paper that introduces new community-wide challenges.

2. Limited Novelty of the Core Finding: The central conclusion—that the robustness of generation-time watermarking is hindered by the difficulty of accurately inverting the carrier signal (in this case, speech tokens)—is not entirely novel to the broader watermarking community. A parallel issue is well-documented in the domain of generative image watermarking, where the imperfect inversion of diffusion models (e.g., DDIM inversion) to recover the initial noise vector is a known cause of poor robustness. The finding that a similar principle applies to the audio domain (inverting audio to tokens) is a logical extension of existing knowledge rather than a surprising or groundbreaking discovery. The proposed solution of reducing information content and using majority voting is also a known technique to mitigate such issues.

**Questions:**

1. Could you clarify if any novel datasets, attack methods, or evaluation metrics were constructed specifically for this benchmark framework? If the contribution is primarily the integration and systematization of existing components, could you further elaborate on why this framework should be considered a new benchmark rather than a toolkit or a comparative study?

2. Based on your extensive evaluation, the paper demonstrates the significant robustness limitations of current generation-time methods. Does this lead you to conclude that post-processing watermarking is the most viable or superior approach for practical, robust audio watermarking in the context of modern TTS systems? What potential research directions could overcome the identified inversion problem for generation-time methods?

---

### Official Review · Reviewer_RdDd · 2025-10-27

**Soundness:** 3
**Presentation:** 3
**Contribution:** 2
**Rating:** 4
**Confidence:** 3

**Summary:**

The paper proposes a benchmark focusing on evaluating how in-process audio-watermarking affects the generated audio quality. The major difference from previous works is that in-process audio-watermarking does not yield a reference audio that can be used to measure similarity scores. The authors use 6 different metrics for reference-free audio-quality benchmarking.

**Strengths:**

- The paper is the first benchmark to evaluate audio quality with in-process watermarking methods.
- The paper conducts extensive experiments.

**Weaknesses:**

The technical contribution of the paper is relatively weak. All the quality metrics are off the shelf and there is no proposed method to improve in-processing watermarked audio quality. The contribution is still solid but might not meet the bar of ICLR.

**Questions:**

Is there any new quality metrics or proposed improvement for in-process watermarking?

---

### Official Review · Reviewer_SRAq · 2025-10-29

**Soundness:** 3
**Presentation:** 3
**Contribution:** 2
**Rating:** 4
**Confidence:** 4

**Summary:**

This paper introduces SpeechWakBench, a large-scale benchmark designed to systematically evaluate the "in-processing" watermarking methods originally developed for text-based LLMs when they are transferred to LLM-based Text-to-Speech (TTS) models. The paper's core contribution is not a new algorithm, but rather this systematic benchmarking framework.
SpeechWakBench provides a comprehensive comparison by:
1. Testing 6 in-processing LLM watermarking methods against 4 post-processing audio watermarking baselines.
2. Running these tests across 3 different state-of-the-art (SOTA) LLM-based TTS models.
3. Introducing a suite of 16 reference-free quality metrics, arguing that this is the only fair way to evaluate in-processing methods which lack a "clean" original audio for comparison.
4. Establishing a unified detectability metric (TPR@X%FPR) to equitably compare different types of watermark detectors (e.g., those outputting p-values vs. those outputting bit error rates).
The paper's main conclusion is that the transferred in-processing watermarking methods "catastrophically fail" in terms of robustness to common audio attacks, performing significantly worse than established post-processing methods. However, the paper does not propose a solution to this problem.

**Strengths:**

1. This paper is the first to systematically investigate the effectiveness and viability of transferring in-processing LLM watermarking techniques (designed for text) to the audio domain via LLM-based TTS models. This is a timely and important research question.
2. The benchmark is extensive, testing 10 different watermarking methods (6 in-processing, 4 post-processing) on 3 SOTA LLM-TTS foundation models, providing a broad overview of the current landscape.

**Weaknesses:**

The paper's primary weakness is its limited scope and analytical depth. While it successfully identifies a significant problem—that a direct, unmodified transfer of in-processing text watermarks to audio fails—it stops there. The "catastrophic failure" in robustness is identified but not adequately analyzed, mitigated, or solved.
1. The analysis of why this failure occurs is superficial. The paper attributes the robustness decline to the "lossy token2wav" process (and the subsequent wav2token reconstruction for detection) without designing experiments to deeply investigate this bottleneck. This is a major missed opportunity.
2. The authors should have explored potential solutions or at least more detailed analyses. For instance:
  - Could the lossy "token-to-waveform-to-token" roundtrip fidelity be improved?
  - Perhaps incorporating a consistency-promoting objective during the speech codec's training could make the tokens more robust to this round-trip conversion?
  - The paper omits testing on flow-matching-based vocoders, which are a mainstream, high-performance approach for token2wav conversion. It is plausible that the robustness problem would be even morepronounced in such models (which lack a simple encoder), a finding that would be critical for a comprehensive benchmark.
3. Without this deeper investigation, the practical utility of in-processing audio watermarking remains highly questionable. The paper's core contribution is presented as the benchmark itself, but its main finding is essentially a negative one (i.e., "this direct transfer doesn't work"). The other "contributions," such as the use of 16 reference-free metrics and the TPR@X%FPR unified metric, are largely standard or common practices in the audio watermarking and speech quality literature, not significant innovations in themselves.
As it stands, the paper benchmarks a failure without providing the community with a path forward, making the contribution feel incomplete.

**Questions:**

1. Beyond identifying the token reconstruction error as the culprit, have the authors attempted any techniques to mitigate this issue?
2. The choice of speech tokenizer seems critical. Different tokenizers will surely have different token-to-wav-to-token reconstruction fidelity. The benchmark would be much stronger if it included a comparative analysis of how different speech tokenizers affect the robustness of in-processing watermarks.
3. Given that flow-matching is a dominant and mainstream method for token2wav synthesis in modern TTS (as noted in your limitations), why was it omitted from the main benchmark? I strongly recommend its inclusion.
4. The paper mentions that the 'X' in TPR@X%FPR is dependent on the 16-bit watermark. A full TPR-FPR (ROC) curve, plotting the trade-off across various detection thresholds, would be far more convincing and provide a more complete picture of detectability than two fixed-point values.
5. The paper does not discuss watermark capacity. What are the capacity requirements for in-processing audio watermarks, and how does performance degrade as the watermark capacity is increased?

---

### Official Review · Reviewer_NEHH · 2025-10-31

**Soundness:** 2
**Presentation:** 3
**Contribution:** 2
**Rating:** 2
**Confidence:** 4

**Summary:**

In this paper, the authors propose a benchmark for the watermarking of audio generated by text-to-speech models. The main novelty of the work is the application of classic "in-gen" text watermarking method to recent TTS model architecture. The stated goal is to answer the followng question:

"**Can the decoder-only watermarking paradigm, successful for text generation by marking discrete tokens, be transferred to TTS systems based on tokenization by LLM?**"

The study comprises an extensive evaluation covering 6 in-processing methods, 4 post-processing baselines, 3 SOTA TTS models, and 10 attacks. The authors main message is a **negative result**: In-gen watermarks, despite mostly conserving  audio quality, *fail catastrophically in terms of robustness* when the audio is subjected to common audio operations(e.g., compression, noise).

The authors then go on to study the robustness of tokenizer itself, demonstrating that token reconstruction from waveform is very brittle, explaining the weakness of "in-gen" watermarking in their setting.

**Strengths:**

**Clarity**: The paper describes every part of the benchmark very well, and should be very replicable.

**Exhaustiveness**: The scope of the paper is impressive. The number of studied metrics, models, and methods studied does provide a good baseline of data for the practitioner.

Sadly, I don't have much else to say: the study truly looks to me like a -- albeit well executed -- data record without many interesting insights.

**Weaknesses:**

**Lack of focus, lack of significance**: All in all, I don't know what the paper wants to be. It's not purely a review o SOTA post-hoc methods since it also studies the applicability of "in-gen" methods. Yet the pitfalls of a naive application of these methods to the audio-gen case are somewhat self-evident to anyone who has designed such methods for text. Robustness of the tokenizer to transformation is paramount -- this has been well-illustrated in a recent work for auto-regressive image models. As such, I am not impressed by the negative result  advertised by the authors: the result is expected. But because of that, the paper, for all its exhaustiveness, seems half-baked: "in-gen"-methods are shown to under-perform, not because of their method but because of the lack of robustness of the tokenizer. Yet a tokenizer is easily changed, fine-tuned, and some strategies can be devised where the embedder/decoder does not need to use the same tokenizer as the generator. Yet the authors do not propose any such thing, giving what I believe an unfair advantage to post-hoc methods which were actually designed for audio, contrary to the in-gen one, whose weakness does not even stem from their design but from the tokenizer's.

**Superiority of in-gen?**: Th authors claim that  there is a "prevailing assumptions about the superiority of in-processing watermarking". I find it hard to believe it to be the case when the recent Synth-ID report [6] basically expound the difficulty of using such methods for image in the real-world.  I don't believe researchers think that In-gen methods have an "intrinsinc" magic that makes them superior. They are attractive because they can, in theory, be readily integrated to the generation model (through some fine-tuning), allowing to share open-source models which produce watermarked content, something impossible with post-hoc methods. The claim of better quality or capacity is known not to be true by default. Taking examples from images, Gaussian-Shading (in-gen) [7] is supposedly lossless in terms of quality but completely non-robust to geometric operations (e.g. crops) whereas the recent post-hoc ChunkySeal [8] boasts the highest capacity of any image watermarking scheme while still being robust to large variety of valuemetric and geometrics operations. All this to say the authors should nuance their claim, because this looks to me as a way to oversell the importance of their main message.


**No concerns about security of post-hoc methods**: The paper advertises itself as a benchmark. As such it should provide a comprehensive review of strengths an weaknesses of current methods. Sadly, the authors falls in the same trap as most watermarking papers by focusing solely on quality and robustness without touching on the question of security. Yet security is one of the three pillar of a production-ready watermarking system and should not be disregarded [1,2]. This is especially important in the audio case since AudioSeal, as been demonstrated to be highly insecure [3], allowing an attacker to easily remove, forge or copy the watermark with minimal effort. The study has not been performed for other post-hoc methods, yet "in-gen" methods should be impervious to such attacks. Since the paper makes the case of the non-robustness of these methods, this should be weighted against the insecurity of post-hoc methods. A simple study of security of other post-hoc methods should be performed for a truly comprehensive benchmark.


**Overselling the metrics**: The author make the use of"TPR@FPR" a contribution of their work. This is such a standard metric that I don't even understand how this can be claimed as a contribution. I find this especially disappointing since, in the case of both post-hoc and "in-gen" methods, a sound $p$-value can be derived, either from the bit-accuracy (using the method in [4] for example) or from a suitable statistical model -- see the extensive study in [5]. This would in my opinion be far better way to unify the robustness metrics, since computing $p$-values allows to **guarantee** a low-FPR without any access to non-watermarked data -- once again see [5].

**Questions:**

- The primary weakness of in-gen methods in the paper stems from the lack of robustness in the current tokenizer/embedder when dealing with corrupted audio. This is a deficiency of the model component, not the watermarking technique itself. I would either appreciate a tentative fix to the tokenizer or a simulated study of the "in-gen" watermark performance when the detector is provided with a certain ratio of "good tokens". This would provide a fairer assessment of the different methods and strengthen the paper in my view.

- The paper focuses exclusively on quality and robustness (against corruption) but completely omits **security** (against removal, forging, or copying attacks). Given that **AudioSeal has known, severe security vulnerabilities**,  could the authors perform a simple security analysis of other post-hoc methods to provide a balanced risk assessment ?

- The authors claim the use of **TPR@FPR** as a contribution, yet deriving sound statistical metrics (p-values)  provides a **guaranteed FPR without requiring access to a separate, large non-watermarked dataset** for calibration. Given that p-values can be readily derived for every studied scheme, **why did the authors choose the empirical TPR@FPR over the more theoretically sound, data-independent p-value approach for unifying your detectability metric?**

**Recommendation**: In its current state, the paper is impressive in its scope but contains many flaws in its experimental design and choices for the reasons I exposed in the weaknesses. As such, I currently recommend rejection. I might increase my score to a borderline accept, provided the authors address my main gripes but I feel like the paper needs much more work to be a valuable benchmark on which practitioner can rely on.

**References**:

-  [1]. Cayre, C. Fontaine and T. Furon, "Watermarking security: theory and practice," in _IEEE Transactions on Signal Processing_, vol. 53, no. 10, pp. 3976-3987, Oct. 2005, doi: 10.1109/TSP.2005.855418.
-  [ 2 ] T. Kalker, "Considerations on watermarking security," _2001 IEEE Fourth Workshop on Multimedia Signal Processing (Cat. No.01TH8564)_, Cannes, France, 2001, pp. 201-206, doi: 10.1109/MMSP.2001.962734.
-  [ 3 ] Bas, Patrick, and Jan Butora. "The AI Waterfall: A Case Study in Integrating Machine Learning and Security." (2025).
- [ 4 ] C. Imadache, E. Giboulot and T. Furon, "Evaluating the security of public surrogate watermark detectors," _ICASSP 2025 - 2025 IEEE International Conference on Acoustics, Speech and Signal Processing (ICASSP)_, Hyderabad, India, 2025, pp. 1-5, doi: 10.1109/ICASSP49660.2025.10889821.

- [ 5 ] Fernandez, Pierre, et al. "Three bricks to consolidate watermarks for large language models." _2023 IEEE international workshop on information forensics and security (WIFS)_. IEEE, 2023.

-  [ 6 ] Gowal, Sven, et al. "SynthID-Image: Image watermarking at internet scale." _arXiv preprint arXiv:2510.09263_ (2025).
- [ 7 ]Yang, Zijin, et al. "Gaussian shading: Provable performance-lossless image watermarking for diffusion models." _Proceedings of the IEEE/CVF Conference on Computer Vision and Pattern Recognition_. 2024.
- [ 8 ] Petrov, Aleksandar, et al. "We Can Hide More Bits: The Unused Watermarking Capacity in Theory and in Practice." _arXiv preprint arXiv:2510.12812_ (2025).

---

### Note · Authors · 2025-12-31

I have read and agree with the venue's withdrawal policy on behalf of myself and my co-authors.